# Serum Albumin in Health and Disease: Esterase, Antioxidant, Transporting and Signaling Properties

**DOI:** 10.3390/ijms221910318

**Published:** 2021-09-25

**Authors:** Daria A. Belinskaia, Polina A. Voronina, Vladimir I. Shmurak, Richard O. Jenkins, Nikolay V. Goncharov

**Affiliations:** 1Sechenov Institute of Evolutionary Physiology and Biochemistry, Russian Academy of Sciences, Thorez 44, 194223 St. Petersburg, Russia; p.a.voron@yandex.ru (P.A.V.); vladimir.shmurak@gmail.com (V.I.S.); ngoncharov@gmail.com (N.V.G.); 2Leicester School of Allied Health Sciences, De Montfort University, The Gateway, Leicester LE1 9BH, UK; roj@dmu.ac.uk

**Keywords:** albumin, esterases, oxidative stress, transport, endothelium, glycocalyx, transcytosis, advanced glycation end products, pathogenesis

## Abstract

Being one of the main proteins in the human body and many animal species, albumin plays a decisive role in the transport of various ions—electrically neutral and charged molecules—and in maintaining the colloidal osmotic pressure of the blood. Albumin is able to bind to almost all known drugs, as well as many nutraceuticals and toxic substances, largely determining their pharmaco- and toxicokinetics. Albumin of humans and respective representatives in cattle and rodents have their own structural features that determine species differences in functional properties. However, albumin is not only passive, but also an active participant of pharmacokinetic and toxicokinetic processes, possessing a number of enzymatic activities. Numerous experiments have shown esterase or pseudoesterase activity of albumin towards a number of endogeneous and exogeneous esters. Due to the free thiol group of Cys34, albumin can serve as a trap for reactive oxygen and nitrogen species, thus participating in redox processes. Glycated albumin makes a significant contribution to the pathogenesis of diabetes and other diseases. The interaction of albumin with blood cells, blood vessels and tissue cells outside the vascular bed is of great importance. Interactions with endothelial glycocalyx and vascular endothelial cells largely determine the integrative role of albumin. This review considers the esterase, antioxidant, transporting and signaling properties of albumin, as well as its structural and functional modifications and their significance in the pathogenesis of certain diseases.

## 1. Introduction: Historical Aspects, Origin and Destination, and Evolutionary and Genetic Features of Albumin

Albumin was probably the first protein that doctors of ancient civilizations paid attention to. Thus, in the 5th century BC, Hippocrates associated kidney disease in his patients with the presence of foamy urine. As we now know, urine foams due to the presence of albumin. The first attempts recorded in historical annals to isolate albumin from urine used vinegar and were undertaken in the 16th century by Paracelsus, but it was not until 1894 that Gourbert first crystallized albumin from horse serum [1]. At first, blood serum was the object of research and a source of albumin, so the definition of “serum albumin” was assigned to the protein. Modern technologies for the isolation of albumin recommend the use of blood plasma as its source [2]. At the same time, the initial reason for the common use of the phrase “serum albumin” was the need to emphasize its difference from egg, milk and plant albumin.

Serum albumin belongs to the albumin superfamily, which also includes vitamin D-binding protein (VDP), alpha-fetoprotein and alpha-albumin (afamin); accordingly, the albumin gene family includes the genes of these four globular proteins [3]. This family is found only in vertebrates [4] and serum albumin is present not only in mammals, but also in birds, some species of frogs, lampreys and salamanders (an exhaustive list is presented on the website albumin.org [5]). In quantitative terms, albumin is the dominant protein in blood plasma or serum and, along with other members of the family, acts as a carrier of endogenous and exogenous substances, including thyroxine, fatty acids (FA) and drugs, while the main “cargo” of VDP is 25-hydroxyvitamin D [3].

All albuminoids are evolutionarily connected with serum albumin [6,7]. It is one of the most evolutionarily changeable proteins; in different species, the differences between albumin domains can reach 70–80%. Clearly, this is due to the development of its special binding characteristics in relation to new ligands over the course of evolution, including hormones, metabolites and toxins. Unlike albumin, differences in the structure of retinol-binding proteins are on average 40%, and less than 10% in the structure of histones [8]. Studies of albuminoid genes have shown that in the FA and thyroxine-binding sites, the contact surface with the neonatal Fc receptor (FcRn), as well as albumin amino acid residues that form a pocket for prostaglandin binding, have been affected by selection to the greatest extent [3]. However, despite the fact that albumin is a rapidly evolving protein, it has two conservative characteristics. First of them is a tertiary structure, which consists mainly of helical regions in the complete absence of any fragments of the beta-sheet. The second one is a pattern of disulfide bonds, of which there are seventeen in the albumin molecule [9]. Due to its presence in all vertebrates, serum albumin can serve as a kind of indicator of the evolutionary stage of a species [10]. Studies of the phylogenetic tree of primate albumin have shown that orangutans were the first to separate from primates; the next were gorillas, later chimpanzees and, finally, humans [11].

The ancestral albumin gene underwent a tripling about 525 million years ago [12], when the first vertebrates appeared. Albumin’s architecture is predominantly spiral and consists of three very similarly shaped domains, which together form a heart shape. However, in the lamprey, which is a “living fossil”, albumin consists of seven domains [13]. Four canonical representatives of the human albumin family are located in tandem in the 4q13.3 region [14]. The *alb* gene of human serum albumin (HSA) consists of 16,961 base pairs from the putative cap site to the first poly(A) site. It is divided into 15 exons, which are symmetrically located in three domains. There are dozens of genetic variants of HSA (see albumin.org [5] for a complete list). The possible effects of some point mutations on the ligand-binding capacity of HSA were investigated in the interactions of five structurally characterized genetic variants of the protein with warfarin, salicylate and diazepam, which are pharmaceuticals with high affinity for albumin [15]. Equilibrium dialysis data revealed a pronounced decrease in high-affinity binding of all three ligands to HSA Canterbury (313Lys→Asn) and HSA Parklands (365Asp→His). For HSA Verona (570Glu→Lys), no change in affinity was found. In the case of HSA Niigata (269Asp→Gly), the affinity was reduced only for salicylate. In the case of HSA Roma (321Glu→Lys), there was decrease in affinity for salicylate and diazepam. In half of the cases, the decrease in the primary association constant reached one order of magnitude, which led to an increase in the unbound fraction of pharmaceuticals by at least 500% at therapeutically relevant molar ratios of the pharmaceutical to the protein. The main reason for the decrease in ligand binding was conformational changes in the region of 313–365, while changes in the charge of the molecule played a secondary role [15].

In humans and many other mammals, the precursor of serum albumin (preproalbumin) has the N-terminal peptide, which is cleaved off before the protein leaves the rough endoplasmic reticulum. The product (proalbumin) is transported to the Golgi apparatus. Limited proteolysis occurs in secretory granules and mature non-glycosylated albumin is secreted into the extracellular environment [1]. Albumin synthesis occurs mainly in hepatocyte polysomes; a healthy adult produces 10–15 g of albumin per day, which is almost 10% of total protein synthesis in the liver [16]. The synthesis of albumin in the liver largely depends on the colloid–osmotic (oncotic) pressure (COP), and its gene expression is regulated according to the principle of feedback [17]. About a third of synthesized albumin remains in plasma, but most of it passes into the extracellular space of muscle tissue and skin. Albumin is mainly lost from the intravascular space by degradation in the skin and muscles. The fate of an albumin molecule, be it degradation, transport across or exchange between pools or compartments, or salvage and recycling, is controlled in large part by its interactions with albumin receptors gp18, gp30, gp60, cubulin, megalin and FcRn [18]. FcRn is widely distributed in many tissues and cell types including vascular, renal (podocytes and the proximal convoluted tubule) and brain endothelia; antigen-presenting cells; and gut, upper airway and alveolar epithelia. The question of whether FcRn could be an efficient transporter of biologics across the nasal epithelial barrier is of particular interest [19]. Also, FcRn is required for the delivery of newly synthesized albumin to the basolateral side of cells and subsequent secretion of albumin into the bloodstream. FcRn is localized mainly within cells and, in addition to IgG, can bind albumin. Lack of FcRn expression in hepatocytes leads to an increase in the level of albumin in bile, its intracellular accumulation and a decrease in the level of circulating albumin [20]. For example, during oncogenesis, cells can lose or suppress FcRn expression. In these cases, cells will not be able to process albumin once it is internalized; instead, it degrades, providing the tumor with nutrients and promoting its growth. Due to its structural features and lack of direct relationship with immune responses, FcRn has been classified as a nonclassical FcγR [21]. IgG and albumin are two major serum proteins that have a relatively long serum half-life, largely due to their interaction with FcRn, which protects them from intracellular degradation through the cellular recycling mechanism.

As for posttranslational modifications, the difference between albumin and other blood proteins is that it is normally not glycosylated (not glycated, if referring to exclusively non-enzymatic glycosylation), although even a small percentage of glycated albumin (GA) makes a significant contribution to the pathogenesis of diabetes mellitus (DM) and other diseases. Lys12, Lys51, Lys199, Lys233, Lys276, Lys281, Lys317, Lys323, Lys439, Lys525, Lys545, Arg10, Arg98, Arg114, Arg160, and Arg428 (Figure 1) are frequent sites of HSA glycation and modification by advanced glycation end-products (AGE) in vivo [22]. Redox modifications of albumin, such as cysteinylation, homocysteinylation and sulfinylation at Cys34, are also known [23]. The albumin molecule contains 17 disulfide bonds and one free thiol group in Cys34 (Figure 1), which determines the participation of albumin in redox reactions.

## 2. Transporting Function and Structural Characteristics of Albumins of Different Species

Albumin can bind various endogenous and exogenous ligands: water and metal cations, FA, hormones, bilirubin, metalloporphyrins, nitric oxide, aspirin, warfarin, ibuprofen, phenylbutazone, etc. [25]. Almost all known drugs and toxic substances are capable of binding with albumin [26]; albumin largely determines their pharmaco- and toxicokinetics, transporting them to target tissues or sites of their biotransformation.

### 2.1. Binding and Transporting Properties of Albumin

Ligands that strongly bind to site I (corresponding to the fatty acid site 7; FA7) are generally believed to be dicarboxylic acids and/or bulky heterocyclic molecules with a negative charge localized to the middle of the molecule, whereas Sudlow’s site II (composed by FA3 and FA4 sites) is preferred by aromatic carboxylates with an extended conformation [27]. Site I is larger than site II, and site I drugs occupy different parts of the binding pocket of subdomain IIA, including the part adjacent to the interface with subdomain IB. This is comprised of two largely apolar clusters with a pair of centrally located polar features, which are formed by the side-chains of Tyr150, His242, Arg257 located at the bottom of the pocket, and Lys195, Lys199, Arg218 and Arg222 on an outer cluster at the pocket entrance. The preference for flat aromatic compounds (such as CMPF; 3-carboxy-4-methyl-5-propyl-2-furanpropionic acid) arises because they are able to fit snugly between the side-chains of Leu238 and Ala291 in the center of the cleft. The enantiomers of warfarin bind in the same position, both being involved in three hydrogen-bonding interactions with Tyr150, His242, and either Lys199 or Arg222 [28]. Thus, site I shows poor stereoselectivity, which might also be due to the flexibility of this site [27]. Site II is a largely apolar cavity with a single dominant polar patch near the pocket entrance, centered on Tyr411 and Arg410. This arrangement of polar and apolar features is consistent with the typical structures of site II drugs, which are aromatic carboxylic acids with a negatively charged acid group at one end of the molecule that is separated by a hydrophobic center [27,29]. Site II is less flexible, as compared to site I, since ligand binding to this site often shows stereoselectivity and is strongly affected by structural modifications of ligands with relatively small groups. For example, (R)-ibuprofen binds to site II with an affinity that is 2.3 times higher than the (S) enantiomer [30]. Some drug binding sites different from sites I and II have been identified in subdomains that are not subdomains IIA or IIIA; the existence of at least one other binding site for probenecid, amitriptyline, debrisoquine and digitoxin was suggested in late 1970s [31,32,33].

Drug binding to HSA can be affected by the presence of other drugs or endogenous compounds, or by the change of HSA structure in certain types of diseased states. A change in the free fraction (fp) may result in altered pharmacokinetics and pharmacodynamics. For example, albumin binding of drugs is decreased in patients with renal diseases such as nephrotic syndrome, chronic renal failure and uremia. In nephrotic syndrome, the albumin concentration drops to 7–25 g/L (normal concentration in adults; 42.0 ± 3.5 g/L) [1,27]. Other factors affecting drug binding, such as accumulation of endogenous inhibitors or carbamylation of albumin, should be taken into consideration. Endogenous inhibitors, such as uremic toxins or fatty acids, are believed to predominantly account for most of the decreased drug binding to albumin [34]. Several anionic uremic toxins responsible for the impaired binding of many drugs to albumin were found in human serum: indoxyl sulfate (IS), indole acetate (IA), hippuric acid (HA) and 3-carboxy-4-methyl-5-propyl-2-furanpropionic acid (CMPF) [35]. Uremic toxins with an indole ring and HA primarily bind to site II, whereas the location of the CMPF-binding site is subdomain IIA, corresponding to site I [36]. Decreased drug binding in liver diseases may be due to a decrease in albumin concentration, the accumulation of endogenous inhibitors (e.g., bilirubin), or changes in albumin structure [27].

The FA1 binding site has evolved to selectively bind to heme with a high affinity, so that HSA participates physiologically in heme scavenging [37]. In turn, heme endows HSA with reactivity and spectroscopic properties similar to those of hemoglobin and myoglobin. Remarkably, both ferric heme (heme-Fe(III)) binding to HSA and human serum heme-albumin (HSA-heme) reactivity are modulated allosterically [37]. A series of compounds, such as chlorpropamide, digitoxin, furosemide, indomethacin, phenylbutazone, sulfisoxazole, tolbutamide and warfarin were reported to allosterically impair peroxynitrite isomerization to nitrate anions (NO_3_^−^) by ferric HSA-heme (HSA-heme-Fe(III)) [38]. The effect of the drugs was ascribed to the pivotal role of Tyr150, a residue that either provides a polar environment in Sudlow’s site I or protrudes into the heme cleft (i.e., the FA1 site).

Flavonoids are plant phenolic secondary metabolites widely distributed in the human diet, and their nutraceutical properties are well recognized nowadays. Values of the dissociation equilibrium constant (*K*_d_) for the binding of flavonoids and related metabolites to Sudlow site I range between 3.3 × 10^−6^ and 3.9 × 10^−5^ M, at pH 7.0 and 20.0 °C, indicating that these flavonoids are mainly bound to HSA in vivo [39]. The protective role of flavonoids, reflecting their transport and storage by HSA, can be inhibited by the increase of plasma FA levels; values of *K*_d_ increase in the presence of saturating amounts of oleate by about two-fold, indicating that FAs act as allosteric inhibitors of flavonoid bioavailability. Therefore, patients affected by metabolic syndrome and characterized by high FA plasma levels may not fully benefit from potential protective effects of dietary flavonoids.

### 2.2. Comparative Characteristics of Human, Bovine and Rat Albumin

When a new drug is being developed, testing for its binding to albumin is a standard procedure. Inexpensive and readily available bovine serum albumin (BSA) is often used in toxicological and pharmacological experiments in vitro as a model of serum albumin [40,41]. *Rattus norvegicus* is one of the main species used in preclinical testing of toxic substances and therapeutic agents in vivo, which is why in some cases, it is necessary to analyze drug binding to rat serum albumin (RSA) [42,43,44,45,46,47]. The end user of the developed drugs is, as a rule, a human being, so the need to include HSA in the panel of a comparative study becomes obvious. A comparative analysis of the structural and functional characteristics of these three albumins is necessary both for improving the methodology of preclinical testing of pharmaceuticals and for identifying features of the evolutionary biochemistry of mammals.

The HSA molecule consists of 585 amino acid residues forming one polypeptide chain. The length of the primary sequence of BSA and RSA is 584 and 583 amino acids, respectively. Their molecular weights based on amino acid composition are 66,439 Da for HSA, 66,267 Da for BSA and 65,871 Da for RSA, but these values may vary due to genetic and post-translational modifications. The amino acid composition of HSA and RSA is 73.0% identical, while that of BSA and RSA is 69.9%. The secondary structure of albumin is comprised of about 67% helical structures and about 33% turns and extended chains [48]. The three-dimensional structure of HSA was obtained rather late, not until the 1990s [49], while a crystal structure of BSA was resolved in 2012 [9,50]. The three-dimensional structure of albumin of rats, which is the main animal species used in pharmacological and toxicological experiments, has not yet been obtained. In the absence of experimental data, the three-dimensional structure of the protein can be constructed using homologous modeling., which is the procedure of building a 3D-model of a protein from its primary sequence and the known three-dimensional structures of homologous proteins [51]. For the first time, using this method on the basis of 14 structures of homologous proteins, we constructed a three-dimensional RSA model [52], which was then tested in in silico experiments [53].

Three homologous domains (Domains I, II, III) consist of two subdomains each (subdomain A containing 6 α-helices and B containing 4 α-helices) and form a quite labile three-dimensional structure of the protein. Domain I of HSA contains 195 amino acid residues, domain II—188 (from 196 to 383) and domain III—202 (from 384 to 585). Domain boundaries are located in the middle of the longest helices (residues 193 and 382). The approximate dimensions of the HSA molecule are 80 × 80 × 30 Å [2,54]. Binding of low-molecular weight ligands occurs at two main sites (Sudlow I site in subdomain IIA and Sudlow II site in subdomain IIIA) and several minor ones (including site III, where bilirubin binds). When albumin interacts with various substances, the effects of cooperativity and allosteric modulation occur, which are usually inherent in multimeric macromolecules [37,55].

Figure 2, Figure 3 and Figure 4 show the three-dimensional structures of sites Sudlow I, Sudlow II and the redox site of HSA, BSA and RSA. Due to the deletion at position 116, amino acid numbering in BSA after amino acid residue 115 is shifted one position relative to the HSA and RSA. In the following description, when it comes to comparing the structure of the enzymes, we give the numbering for HSA as a reference, and if necessary, the corresponding amino acid number in BSA is given in brackets, for example, Tyr150 (Tyr149).

It can be seen from Figure 2 and Figure 3 that Sudlow site I is less conservative than Sudlow site II; thus, Lys195 and Lys199 in HSA are replaced by more branched arginines Arg194 and Arg198 in BSA. In RSA, Lys195 is also replaced by arginine. Arg222 in HSA and RSA are replaced by Lys221 in BSA. Leu219(Leu218) in HSA and BSA is replaced by Met219 in RSA. Isoleucine Ile264(Ile263) in HSA and BSA correspond to Met264 in RSA. There is isoleucine at position 290(289) in HSA and BSA, and in RSA it is replaced by leucine. Valine at position 293(292) in HSA and BSA is replaced by isoleucine in RSA. Histidines His242(His241) and His288(His287) in the primary sequence of HSA and BSA are replaced by Asn242 and Gln288 in the RSA molecule. The latter substitutions are of particular interest, since His242(His241) and His288(His287) are located not far from Tyr150(Tyr149), which is supposed to play a major role in the esterase activity of albumin. According to our computational experiments on interactions of albumin with organophosphates (OP) [56,57,58,59], the imidazole ring of His242(241) can pull off the proton of the hydroxyl group Tyr150(Tyr149) and thus regulate the hydrolytic activity of tyrosine. It should be expected that interspecies differences in the binding and catalytic properties of albumin will be correlated with differences in the characteristics of Sudlow site I.

Sudlow site II is much more conservative than Sudlow I (Figure 3). There are substitutions only at positions 388 (Ile388, Ile387 and Val388 in HSA, BSA and RSA), 390 (Gln390, Gln389 and Thr390 in HSA, BSA and RSA), 407 (Leu407, Leu406 and Ile407 in HSA, BSA and RSA) and 449 (Ala449, Thr448 and Val449 in HSA, BSA and RSA). All replacements, except for the homologous substitution at position 407, are located at a distance not influencing the catalytic tyrosine Tyr411(Tyr410).

Also, considerable differences can be observed in the structure of the redox site (Figure 4). Gln33, Phe36, and Thr83 in HSA and BSA are replaced by Lys33, Tyr36 and Asn83 in RSA, respectively. But most notably, Tyr140(139) in HSA and BSA has been replaced by His140 in RSA.

We have previously shown that Sudlow I and Cys34 sites in HSA and BSA molecules mutually influence each other; a change in the conformation of one leads to conformational changes in the other [60,61]. At the same time, in the redox site, the main role in reciprocal regulation is played by amino acid residues Cys34, His39 and Tyr140(139), and their mutual arrangement; the position of the thiol group of the cysteine and the hydroxyl group of the tyrosine relative to the imidazole ring of His39. How this system works in RSA, where Tyr140(139) is replaced by His140, and how such a replacement affects the behavior and availability of Cys34 is still unknown. It is even possible that in rats this mechanism is more perfect than in humans and cattle, since rodents are more omnivorous and better adapted to the environment.

The structural differences in HSA, BSA and RSA determine differences in the equilibrium and kinetic constants of their esterase activity. For example, in NPA and paraoxon, for the first time, we carried out a comparative analysis of esterase and paraoxonase activities of HSA, BSA and RSA in vitro [62]. According to the data obtained, Sudlow site I of RSA has the highest catalytic efficiency towards NPA, which is approximately two times higher than that of HSA and 5.5 times higher than the catalytic efficiency of BSA. These differences are due exclusively to the unequal affinity of Sudlow site I in the studied albumins for the substrate, because the number of turnovers is the same for HSA, BSA and RSA. The difference in the dissociation constants of RSA and HSA in Sudlow site II is insignificant, while this parameter in BSA is two times lower, which is also due to the higher affinity of Sudlow site II in BSA site for the substrate. Similar results were obtained for paraoxon. The data obtained indicate that the equilibrium and kinetic characteristics of both Sudlow sites of HSA and RSA differ from each other to a lesser extent compared to the characteristics of these sites in BSA. The characteristics of Sudlow II in HSA and RSA are practically the same. The latter finding confirms the assumption concerning the evolutionary conservatism of Sudlow site II in comparison with Sudlow I.

### 2.3. Albumins of Other Species

In addition to HSA and BSA, three-dimensional structures of leporine (LSA), equine (ESA) [9], caprine (CapSA) and ovine (OSA) [63] serum albumins, as well as structures of recombinant canine (rCanSA) [64] and feline albumins (rFSA) [65], have been obtained so far. It is worth mentioning the main differences found when comparing these albumins.

The ligand-binding pockets in BSA, ESA and LSA [9] revealed different amino acid compositions and conformations in comparison to HSA in some cases; however, much more significant differences were observed on the surface of the molecules. The hydrophobic residues located at the bottom of Sudlow sites are more highly conserved than the polar residues located at their entrances. These changes in protein sequence adjust the shape and charge distribution of the pockets and modulate the affinity of particular albumins for selective ligands [9].

A comparison of OSA and CapSA with the closely related BSA revealed that, despite 98% sequence similarity, OSA binds only two molecules of 3,5-diiodosalicylic acid (DIS), whereas CapSA binds six molecules of this ligand [63]. In BSA, DIS molecules are bound at four positions. Two additional binding positions for DIS in BSA are located in FA-binding site 1 in the IB domain and in Sudlow site II between domains II and III. In CapSA, two additional locations (absent in BSA) for binding of DIS are observed. They are localized in niches on the surfaces of domains I and III. Additionally, analysis of the electrostatic surface potential of serum albumins revealed some differences in the distribution of positive and negative charges, which means that interactions with other proteins found in nature, especially with antibodies, may be different for albumins from various species [63].

The environment around the Cys34 in rCanSA and rFSA is more polar and flexible compared to HSA, which explains why the free sulfhydryl group ratio of Cys34 is lower in cat and dog albumins compared to HSA [64,65]. Interestingly, warfarin and phenylbutazone (ligands of Sudlow site I) cannot bind to canine albumin, though superposition of rCanSA and HSA shows that the architectures of their Sudlow sites are identical and the side chains of Tyr150, Lys199, and Arg222 are located at the same positions [64]. The surface charge distribution might be responsible for these binding features.

## 3. Enzymatic Activity of Albumin

Albumin is not just passive, but an active participant in pharmacokinetic and toxicokinetic processes too. The esterase or pseudoesterase activity of albumin against α-naphthyl acetate and *p*-nitrophenyl acetate (NPA), FA esters, aspirin, ketoprofen glucuronide, cyclophosphamide, nicotinic acid esters, octanoyl-ghrelin, nitroacetanilide, *p*-nitrofluoroacetanilide and some OPs have been demonstrated in numerous experiments [56,66,67,68]. A typical example of the pseudoesterase activity of albumin is acetylation. In this reaction, the substrate is consumed through the formation of covalent bonds, with the participation of many amino acids in the albumin molecule. It was shown that 82 amino acid residues of HSA can be acetylated when interacting with NPA, including 59 lysines, 10 serines, 8 threonines, 4 tyrosines and one aspartate [69].

The phosphatase activity of albumin is of particular interest. It includes the activity of phosphomonoesterase (EC 3.1.3...) [70], RNA hydrolase or phosphodiesterase (EC 3.1.4.16) [71] and phosphotriesterase (EC 3.1.8.1 and 3.1.8.2) [72,73]. Subclass 3.1.8 (phosphotriester hydrolases) includes aryldialkyl phosphatase (EC 3.1.8.1) and diisopropyl fluorophosphatase (EC 3.1.8.2) [74,75]. Aryldialkyl phosphatase is known as paraoxonase 1, which hydrolyzes the esters of tribasic phosphoric acid, dibasic phosphonic acid and monobasic phosphinic acid. It is known that PON1 activity requires divalent cations (mainly calcium ones) [76]. The fundamental difference between the analogous activity of albumin is calcium independence, which is taken into consideration in the differential analysis of the activity of these enzymes [73,77]. In toxicology, understanding the mechanisms of interaction between OPs and albumin can contribute to the development of adjuvant therapy [59]. Other manifestations of serum albumin enzymatic activity include those associated with the metabolism of prostanoids (prostaglandin D-synthase and others) [78,79,80]. Glucuronidase [81,82] and enolase [83] activities are quite unusual in albumin, however the latter can be used for differential diagnosis of benign and malignant tumors.

BSA and HSA catalyze the aldol reaction of aromatic aldehydes and acetone with saturation kinetics and moderate stereoselectivity; the reaction occurs in domain IIA and is inhibited by warfarin [84], which suggests the involvement of the Sudlow I site. Luisi et al. [85] identified a 101 amino acid polypeptide derived from the sequence of domain IIA HSA. The peptide contains eight cysteine residues, which form disulfide bonds, stabilizing its structure [85]. This polypeptide retains the ability to bind typical ligands of domain IIA, such as warfarin and efavirenz, and also has aldolase activity and the ability to regulate stereoselective diketone reduction. It has been suggested that some simple reactions catalyzed by serum albumin with Michaelis–Menten kinetics involve nonspecific substrate binding and catalysis by local functional groups [86]. These various active sites can nonselectively interact with many hydrophobic negatively charged ligands, with the lysine residue acting as a primitive active site facilitating this nonselective interaction [87]. A method has been proposed for predicting promiscuous enzyme activity using a graphical representation known as a molecular signature; enolase activity was one of the first promiscuous activities reported in literature [88]. However, in the presence of a denaturing agent such as urea, the hydrophobic interaction of the substrate with the binding site gives way to electrostatic interactions [89].

Most enzymes are capable of catalyzing physiologically irrelevant (secondary, “promiscuous”) reactions in addition to those that have become primary for them as a result of evolution. After a detailed examination of the issue, the number of “promiscuous” reactions turned out to be quite large [90,91], so it is the rule rather than the exception. However, the catalytic “promiscuity” of albumin, in our opinion, arose as a result of the loss (and not the acquisition) of some specialized activities, for example the activity of esterases (hydrolases) in digestive functions.

Kemp’s elimination, which is a prototypical reaction for the elimination of a proton from carbon, plays a certain role in the mechanism of albumin “promiscuity”. The reaction takes place in the Stern layer, at the interface between the micelle head or the surface of the protein and water, so that a significant acceleration of the reaction can be achieved regardless of the spatial arrangement of the substrate [92,93]. It should be noted that the reaction rate is reduced in protic solvents such as water compared to aprotic organic solvents. The electrostatic component of hydrogen bonds is the main factor in the inhibitory effects of water, while an external electric field oriented in the direction of the charge transfer increases the reaction rate [94]. On the other hand, the mechanism of Kemp elimination in protein molecules is associated with the presence of aromatic amino acid residues (Trp, Tyr, Phe), providing stacking interaction with hydrogen bond donors (Lys, Arg, Ser, Tyr, His, water molecules) [95]. We have previously proposed an explanation for albumin-mediated hydrolysis of some substrates by the existence of catalytic dyads (in contrast to the catalytic triads in cholinesterases) His-Tyr or Lys-Tyr, in which histidine or lysine residues function as acid residues and proton donors, and the tyrosine residue is a catalytic base [66].

It was in 1986 when concern was expressed for the first time that the classification of esterases did not take into account the facts that had accumulated by that time. Moreover, it was albumin that was cited as an example of a protein that exhibits esterase activity, but it does not appear in the existing classification [96]. Our own experiments and literature data from recent years emphasize the importance of assessing the enzymatic activity of albumin for the purposes of pharmaco- and toxicokinetics [57,58,66,97]. Substrate nonspecificity (promiscuity) and the absence of dependence on Ca^2+^ ions do not allow the identification of albumin as an enzyme, but paradoxically it is precisely these features of that can allow it to find its place in the nomenclature.

In living systems, unlike in in vitro experiments, biochemical processes take place in a medium containing high concentrations of macromolecules (50–400 mg/mL). In human blood plasma, the density of protein macromolecules normally reaches 90 mg/mL; such conditions are called molecular crowding. Due to the dense medium, the volume of the available solvent decreases, which can cause pathological aggregation and affect the protein structure, folding, shape, conformational stability, enzymatic activity, the binding of small molecules, and the interaction of proteins with each other and with nucleic acids [98].

The crowding effect was also demonstrated for albumin with the help of Raman spectroscopy; a dense medium affects the strength of intramolecular hydrogen bonds in the BSA molecule, which forces the protein molecule to acquire a more compact structure [99]. Zhu et al. showed that molecular crowding has an influence on the binding of saturated medium-chain FAs and unsaturated long-chain FAs to BSA [100]. Thus, it can be assumed that the characteristics of the binding and esterase activities of albumin in the bloodstream will differ from the constants measured in the “ideal solution” in in vitro and in silico experiments. In this regard, we emphasize the importance of developing test systems that simulate the activity of albumin under molecular crowding conditions.

## 4. Redox Modulation and Redox Activity of Albumin

### 4.1. Antioxidant Properties of Albumin

The redox status of the thiol group of the Cys34 residue ensures the heterogeneity of albumin isoforms: mercaptalbumin (reduced albumin, HMA) and non-mercaptalbumin-1 and -2 (oxidized albumin variants HNA-1 and HNA-2) [101]. HNA-1 is a mixed disulfide with Cys or cysteinylglycine (CysGly), and to a lesser extent, homocysteine (HCys) or glutathione (GSH) residues; albumin with a cysteine residue oxidized to sulfinic or sulfonic acid is named HNA-2. Of the total amount of albumin, healthy young people have 70–80% as HMA, 20–30% as HNA-1, while only 2 to 5% is HNA-2 [102]. Oxidized forms of albumin differ in physical and chemical properties from the reduced forms. Thus, an increase of the COP of oxidized albumin was shown in in vitro experiments using hypochlorite, and it was also found in patients with chronic kidney disease [103]. An affinity for endogenous ligands such as bilirubin and tryptophan, as well as for exogenous pharmaceuticals like warfarin and diazepam, decreases proportionally to the level of oxidized albumin [104]. The lipid affinity differs too, for example, pro-atherosclerotic lysophosphatidylcholine and lysophosphatidic acid have a higher affinity for the oxidized isoform, whereas anti-atherosclerotic derivatives of eicosapentaenoic and docosahexaenoic acids have a higher affinity for the reduced albumin isoform [105].

HSA residues, that are very susceptible to oxidation, are Cys34 primarily, but also have residues on tyrosine Tyr84, 138, 140, 161, 263, 319, 332, 334, 353 and 370, on methionine Met87 and Met123, and on tryptophan Trp214 [106]. In albumin in healthy human plasma, about 80% of all thiols are Cys34 residues [107]. Oxidization to sulfenic acid (HSA-SOH) inactivates hydrogen peroxide stoichiometrically, as well as peroxynitrite, superoxide anions and hypochlorous acid [108,109]. When oxidative stress caused by ROS occurs, Cys34 forms a disulfide with free cysteine or glutathione; oxidation changes the three-dimensional structure of HSA and affects the binding of many xenobiotics (pharmaceuticals and toxic substances).

It appears that FAs play an important role in the regulation of albumin antioxidant properties [110]. The binding of FAs by albumin changes the conformation of Sudlow I and II sites and increases the fluorescence quantum yield of dansylamide (ligand of the Sudlow I site) and dansylsarcosine (ligand of the Sudlow II site); in addition, FAs increased the steric availability of the thiol group Cys34 and stimulated its reactivity with respect to 5,5′-dithiobis-2-nitrobenzoic acid (DTNB). Thus, the binding of FAs suggests isochronic regulation of two important functions of the protein: transportation and antioxidation [110]. Moreover, albumin potentiates the antioxidant status of the organism through the binding of bilirubin (Site III ligand [111]) and polyunsaturated FAs that interact with Arg117, Lys351 and Lys475 residues [108].

Binding of polyvalent metal ions by albumin plays a significant role in its redox properties. Thus, albumin takes part in the transport of copper [112]. Copper is a cofactor of many enzymes and a participant in redox reactions and signaling pathways in the body in normal and pathological states [113]. The main binding site of Cu(II) cations is the N-terminal site (NTS) of HSA Asp-Ala-His-Lys [114]. It appears that there is a binding site for Cu (I) in the albumin structure. Using spectroscopic and computational methods, it was shown that the imidazole rings of two histidines play a key role in the binding of the Cu(I) cation [115]. The N-terminal region in complex copper ions has superoxide dismutase activity [116]. In addition, the prooxidant properties of albumin should be noted. Thus, Cu^2+^ ions associated with albumin potentiate the formation of ascorbate radicals, followed by oxidation of formed Cu^+^ ions with molecular oxygen and protons back to Cu^2+^ [110].

The list of albumin activities associated with redox-modulation of blood plasma and intercellular fluid includes some kinds of enzyme activities such as thioesterase [117,118], glutathione peroxidases and cysteine peroxidases, as well as the activity of peroxidase against lipid hydroperoxides [119,120,121]. An important role of two albumin cysteine residues, Cys392 and Cys438, which form redox-sensitive disulfides in the complex of albumin with palmitoyl-CoA, should be noted [121]. Cys34 is one of the most important “scavengers” of reactive oxygen species (ROS), although six methionine residues also contribute to the antioxidant properties of albumin [108,122]. The residues of Met87 and Met123 are usually oxidized to methionine sulfoxide, especially in renal failure and DM.

### 4.2. Practical Aspects of Redox Status of Albumin

Levels of oxidized albumin are an oxidative stress marker. Thus, the level of Cys34-cysteinylated albumin is significantly increased in patients who suffer from DM, liver and kidney diseases [104]. Moreover, levels of oxidized albumin are a biomarker of the severity of oxidative stress in such diseases as Duchenne muscular dystrophy [123], Alzheimer and Parkinson’s disease [124,125], hyperparathyroidism [126] and acute ischemic stroke [127]. An assessment of Japanese residents showed that atherosclerotic changes in the carotid artery are inversely correlated with the level of HMA in blood plasma [128]. Excessive formation of ROS contributes to oxidative damage, inflammation, endothelial dysfunction and fibrosis of the renal tissue in kidney disease [129]. Patients with DM often have chronic kidney disease (CKD), so the main cause of kidney damage is considered to be GA, although a comparative analysis of glycated and oxidized albumin has not yet been made. Negative effects of albumin, both modified and non-modified, on the epithelial cells of the tubules have not been studied enough. It is suggested that albumin oxidation precedes or occurs in the early stages of CKD. Healthy human albumin overcomes the glomerular filtration barrier, but is reabsorbed by receptor-mediated endocytosis in proximal (71%) and distal tubule cells (26%) [106]. Albumin binds to the receptor of the megalin–cubulin complex and is directed to clathrin-coated vesicles; then, endocytosis and acidification of endosomes occurs, causing dissociation of albumin from the megalin–cubulin complex and binding of albumin to FcRn. Then, the albumin is either transposed to the lysosomes or returns to the blood via the transcytose route, while the receptors are exposed to recycling. However, levels of oxidized albumin correlate with decreases in the glomerular filtration rate. Oxidized albumin directly effects neutrophils by increasing lipocalin levels associated with neutrophil gelatinase, which are a generally accepted biomarker of renal damage in patients and under various experimental conditions. Moreover, oxidized albumin in patients with CKD independently correlates with higher plasma levels of proinflammatory cytokines TGF-β1, TNF-α, IL-1β and IL-6 [106].

It is assumed that both HNA-1 and HNA-2 cause the progression of inflammatory processes, which is associated with an increase in the level of pro-inflammatory cytokines and markers of damage to certain tissues and organs. According to recent studies, we can suggest that oxidized albumin isoforms are independent pathogenetic factors of many common and socially significant diseases, and their levels are closely related to the state of human nutrition [101]. However, in many respects, the specificity of responses in different tissues to the effects of different forms of modified albumin remains unclear.

## 5. GA: Biomarker and Pathogenetic Factors of DM

### 5.1. The Role of AGEs and GA in DM Pathophysiology

DM is a global epidemic that is growing at an alarming rate and is associated with increasing population mortality rates. In 2018, there were 34.2 million people with DM in the United States alone. Globally, there are 425 million people, and this number is expected to rise to 629 million by 2045 [130]. Gestational DM affects 2% to 14% of pregnant women in the United States each year [131]. In individuals with DM, there is an increase in the process of non-enzymatic condensation of sugars with nucleic acids, proteins and lipids, which ultimately forms AGEs. Glycation changes their structure and function, which leads to dysfunction in cells and cytotoxic effects, because of which, AGEs are called glycotoxins. AGEs can arise from physiological processes when not balanced by detoxification mechanisms, or from external sources such as food, cigarette smoke and air pollution. Their accumulation leads to inflammation and oxidative stress, mainly through the activation of specific AGE receptors (RAGEs; Figure 5) [132].

RAGEs were first described in 1992; later, other AGE receptors were identified: AGER-1, 2, 3, as well as the scavenger receptor CD36 [133,134]. RAGE and AGER-2 are involved in the initiation of inflammatory processes, while the AGER-1, -3 and CD36 receptors are responsible for the detoxification of AGEs [132,135,136]. RAGEs are multiligand binding receptors belonging to the immunoglobulin superfamily; their ligand-binding domain recognizes—in addition to AGEs—a large number of molecules, among which the most important are S100 proteins, high mobility group box-1 protein (HMGB1), β-amyloid and macrophage-1 antigens (Mac-1) [137]. Thus, RAGEs act as nonspecific pattern recognition receptors that can function as environmental sensors [132]. Glycated foods and RAGE activation are associated with the pathophysiology of many metabolic diseases such as type 2 DM (DM2), food allergies, asthma, chronic obstructive pulmonary disease (COPD), acute renal failure (ARF), Alzheimer’s disease and polycystic ovary syndrome (PCOS) [138,139,140]. AGEs promote carcinogenesis in chronic local inflammation induced by *Helicobacter pylori* [141].

Signaling pathways in RAGE activation include p21ras, MAP kinases, Rho GTPases, c-Jun N-terminal kinase (JNK) and JAK/STAT, which lead to the migration of transcription factors into the nucleus and the expression of genes that regulate chemotaxis, cell activation and proliferation (Figure 5) [132]. NF-κB, NFAT, STAT, AP-1, ERK1/2 and CREB-TF (CREB, cAMP response element-binding protein) bind to their specific promoters for the transcription of genes encoding pro-inflammatory cytokines (e.g., IL-1, IL-2 and IL-4), proapoptotic proteins (e.g., p53-Bax, which initiates the caspase cascade), and surface proteins such as EC adhesion molecules [142,143,144]. RAGEs are constitutively expressed in many tissues, while RAGE hyperactivation induces stimulation of the PI3K–PKB–IKK pathway, resulting in NF-κB binding to RAGE promoters and autoamplification of expression (Figure 5) [132]. Another self-amplification loop is that AGEs induce oxidative stress through RAGE activation, followed by NADPH oxidase hyperactivation, ROS generation, increased AGEs, and increased RAGE expression (Figure 5) [145]. RAGE expression can be increased not only due to AGE exposure, but also due to exposure to proinflammatory cytokines [146]. AGE-activated ECs “attract” lymphocytes and delay apoptosis of monocytes, increasing the duration of inflammation [147].

Modifications of plasma proteins, structural proteins and other macromolecules are enhanced in DM not only due to increased glycation (secondary to increased glucose concentrations), but also due to oxidative stress that occurs during the course of the disease, and may be a sign of uncomplicated DM. The combined effects of glycation and oxidation can accelerate the development of comorbidities in DM. While glucose itself contains a carbonyl group that participates in the initial glycation reaction, the most important and reactive carbonyls are formed by oxidative reactions that damage either carbohydrates (including glucose itself) or lipids. The resulting carbonyl-containing intermediates then modify the proteins to give “glyoxidation” and “lipoxidation” products, respectively. This common pathway of glucose and lipid-mediated stress is the basis of the carbonyl stress hypothesis [148].

### 5.2. GA as a Diagnostic Tool for DM

AGEs are formed endogenously even in healthy individuals, but recent studies have shown that diet is an important exogenous source of AGEs [149,150]. Since the targets of glycation are free amino groups, potentially any protein can be modified, and every tissue can accumulate glycotoxins. In terms of AGE accumulation and duration, long-living proteins are the most important for study; these are components of the extracellular matrix (collagen, laminin and elastin), lens protein α-crystallin, cartilage and hemoglobin [132,142]. For example, glucose and glycated hemoglobin A1c (HbA1c) are currently standard biomarkers for monitoring DM. HbA1c is representative of glycemic data for the previous 2–3 months but is not an adequate marker for monitoring ongoing therapy. In mild to moderate gestational DM, when glucose levels can change significantly even during the day, the use of glycated hemoglobin as a marker can disorient the patient and even the doctor [131]. In addition, under certain conditions, the measurement of HbA1c is unreliable, in particular, for patients with modified red blood cells or renal failure [151,152]. Thus, there is a need for an intermediate biomarker that can be effectively used to monitor the glycemic status of patients. Albumin, which has a relatively short half-life period (20–21 days), can serve as such a biomarker. Albumin, the amount of which significantly exceeds the amount of other blood plasma proteins, is susceptible to glycation in the first place and allows prediction of the risk of developing DM even in the case of euglycemia [153]. One of the main forms of AGE-albumin found in vivo is carboxymethyl lysine (CML). Others that should be noted are glyoxal (GO), methylglyoxal (MG) and 3-deoxyglucosone (3-DG) [132,154].

As already mentioned, albumin is synthesized and enters the bloodstream as a non-glycosylated protein, but even in a healthy persons’ blood plasma, a certain proportion of albumin molecules undergo glycation over time, which can affect their structural and functional characteristics [155,156]. According to some reports, in normoglycemic blood 10 to 18% of circulating proteins are glycated in vivo, while in diabetic blood this proportion reaches up to 40% [131,157]. Comparative analysis of various sugars’ ability to interact with BSA in vitro showed that D-galactose is more reactive than D-glucose or D-lactose, although only albumin-lactose conjugates were recognized by specific lectins [158]. So far, more than 60 sites of glycation have been identified on HSA. The most accessible lysines for conjugation are Lys256 and Lys420, although Lys525 is considered the most reactive in HSA [159,160]. Apart from lysines, the most reactive amino acid residues are arginine and histidine [132]. Oxidized albumin is easier to glycate, even at physiological glucose concentration (5 mM) [157]. The accumulation of AGEs changes the structure and function of proteins, turning them into potential targets of the immune system, which can result in the production of autoantibodies against AGEs. Albumin was no exception; HSA antibodies were found in the blood of patients with atherosclerotic vascular injury and signs of DM [161]. Moreover, IgG and IgA autoantibodies to HSA may have diagnostic value in autoimmune bullous dermatoses (ABD), while cutaneous autoantigens have not yet been identified [162]. The phospho-p38 signaling pathway, associated with DNA damage, is a potential target for the treatment of patients with ABD-positive serum autoantibodies to HSA.

Glycated albumin is used both for diagnostics and for experimental study of the systemic effects of AGEs, many of which are mediated by NF-κB, a universal transcription factor that controls the expression of genes in the immune response, apoptosis, and cell cycle [163]. As already noted, plasma glucose and HbA1c are currently generally recognized as markers of DM. Fructosamine has become an alternative to HbA1c as a DM marker and plays an increasingly important role in the diagnosis of DM. It is considered a measure of glycation of circulating proteins, the main component of which—in terms of availability and assessment of the scale of observed changes—is albumin [130]. Blood plasma albumin is directly exposed to circulating glucose, so it is gradually replacing HbA1c in glycemic monitoring of patients with DM [151]. However, the accuracy of determining the degree of hyperglycemia by fructosamine is relatively low, not only because different proteins interact differently with glucose and other sugars, but also because bilirubin, uric acid and a number of other low molecular weight compounds contribute to method errors. Other disadvantages of the test include the lack of a generally accepted standard, and even its low availability [164]. It cannot be said that various methods for determination of glycated albumin are simple and accessible; we note here ion-exchange HPLC, boronate affinity chromatography, immunoassays (radioimmunoassay and enzyme-linked immunosorbent assay), colorimetric methods with thiobarbituric acid, and enzymatic methods using protein oxidase and ketamine oxidase. Nevertheless, in recent years, the “Lucica GA-LR” enzymatic method (Asahi Kasei Pharma Corporation, Japan), which has high reproducibility, accuracy and a good correlation with A1C has become the most popular [164]. A test system on an indicator strip has been developed to measure the proportion of glycated albumin in total serum albumin. Aptamers with gold nanoparticles were used to carry out colorimetric measurements. Both glycated and non-glycated albumin can be measured in the appropriate physiological concentration ranges—from 50 μM to 300 μM—with a limit of detection (LoD) of 6.5 μM for glycated albumin, and from 500 μM to 750 μM with an LoD of 21 μM for non-glycated albumin [131].

The advantage of GA measurement in clinical practice is its versatility, both as a mediator of inflammation and as a marker of hyperglycemia. A deeper understanding of GA’s role may lead to its acceptance as an independent marker of the inflammatory process [165]. GA allows prediction of the risk of death in dialysis patients with DM [151], and in combination with hsCRP, maximizes the accuracy of predicting cardiovascular diseases, especially those associated with left ventricular hypertrophy in patients with diabetic CKD [152]. It should be especially noted that glycated albumin can be transformed into amyloid fibrils rich in β-layers [166].

### 5.3. Species Differences in Glycation Properties of Albumin

There are contradictory data on the effects of glycation on the antioxidant properties of albumin as well as on the effects of Cys34 oxidation on the binding activity of Sudlow sites [167,168,169,170], which may be due to interspecies differences, the nature and concentration of carbohydrates involved in the reaction (glucose, methylglyoxal), and the conditions of incubation with monosaccharides [155]. Differences between HSA and BSA are of particular interest; glycation of HSA dramatically reduces its antioxidant activity, while glycation of BSA somewhat enhances its antioxidant properties. These data correlate with the results of computational experiments aimed at studying the effects of the redox status of HSA and BSA on their binding and esterase activities towards paraoxon [60,61]. From an evolutionary and comparative biochemistry/physiology point of view, the following fact seems to be important: the concentration of glucose in the blood plasma of birds is 1.5–2 times higher than in mammals of a similar mass (so-called benign hyperglycemia), but avian albumin (for example, CSA—Chicken Serum Albumin) gets glycated to a lesser extent than BSA, even when albumins have been exposed to increasing glucose concentrations of up to 500 mM in in vitro experiments [171]. Analysis of protein structures suggests that the relative resistance of CSA to glycation may be associated with fewer lysine residues and variations in protein folding that protect lysine residues from interacting with plasma glucose. Comparative analysis of reconstructed albumin sequences indicates that the ancestors of birds had 6-8 fewer lysine residues in the albumin molecule compared to mammalian albumin [171]. Benign hyperglycemia is a common physiological feature of birds, and the development of mechanisms to resist albumin glycation is apparently inextricably linked with their evolution. It is believed that the development of benign hyperglycemia in birds coincided with a radical restructuring of their genome, which resulted in the loss of important genes, including the gene encoding GLUT4, a transporter responsible for insulin-dependent glucose transport in insulin-sensitive cells of other vertebrates. This loss appears to have led to the remodeling of the insulin-dependent signaling pathway in avian tissues [172].

## 6. Interaction of Albumin with EC: Glycocalyx, Transcytosis and Glycoprotein CD36

AGEs cause multiple metabolic disturbances in the vascular wall and can lead to endothelial dysfunction. Since a significant proportion of AGEs are represented by glycated albumin, it is necessary to consider the features of albumin interactions with the vascular endothelium. Albumin interacts with ECs through extracellular molecular representatives, among which there are receptors, but most of which are combined into a glycocalyx—a dynamic and heterogeneously composed “layer” between EC membranes—on the one hand, and with blood components (plasma and blood cells) on the other hand. The endothelial glycocalyx is a layer of glycoproteins associated with EC membranes, which retains 700 to 1000 mL of practically non-circulating plasma volume in the human body. This intravascular layer maintains its own COP due to the plasma proteins it contains (primarily albumin), which are retained inside the endothelial glycocalyx (endothelial glycocalyx layer, EGL). Consequently, it has a higher COP than circulating plasma [173]. According to some estimates, EGL provides approximately 60% of intravascular COP [174]. Structurally, EGL is a negatively charged gel-like layer that includes oligosaccharide and polysaccharide chains of glycosaminoglycans (heparan sulfate, chondroitin sulfate, dermatan sulfate, keratan sulfate, hyaluronan), which are covalently bound to glycoproteins and proteoglycans—both membrane (antithrombin III, integrins, selectins, syndecans, glypicans) and soluble (perlecan, biglycan, versican, decorin, mimecan). The EGL has a thickness of 0.1 to 4.5 μm, depending on the location and size of the vessel, and serves as a kind of reservoir of proteins and polysaccharides such as antithrombin III and heparan sulfate [173].

An intact EGL maintains separation between circulating plasma and the vascular endothelium, creating an exclusion zone that prevents blood cells from contacting the EC surface. In the presence of an intact EGL, water and electrolytes freely pass first through this layer and then outside the EC through intercellular clefts. This “exclusion zone” also prevents contact with the endothelium of high molecular weight colloids > 70 kDa. Albumin is practically the only plasma protein that easily moves between blood plasma and the EGL due to the selective permeability of the EGL for natural colloids with molecular weights < 70 kDa [175]. A schematic illustration of the space between plasma and the EGL with albumin molecules is shown in Figure 6.

The molecular radius of albumin prevents its passage between adjacent cells of the intact endothelial monolayer, which limits paracellular diffusion for molecules less than 3–5 nm in size [176,177,178]. However, the detection of albumin in interstitial and lymph fluids—up to 40–60% of the plasma level—indicates that the protein is able to leave the microvessel lumen even in the absence of inflammation [179,180]. Milici et al. [181] demonstrated that albumin injected into the bloodstream of animals was later found in the intracellular vesicles of the capillary endothelium; in some cases, these vesicles released albumin into the interstitium, although no rupture of interendothelial contacts was ever observed. This vesicular transport is known as transcytosis and is mediated by the caveolae; knockdown or deficiency of caveolin-1, without which the caveolae cannot exist, prevents albumin transcytosis [182,183]. Thus, under physiological conditions, albumin transcytosis is the main route of albumin transfer from the bloodstream to the interstitium [181]. Given the important role of albumin in the transport of drugs and toxic substances, the relevance of studying the kinetics and mechanisms of albumin transcytosis in blood vessels can hardly be overestimated [184,185]. Little is known about the regulation of transendothelial transport of albumin, which is partly due to technical difficulties associated with its study—that is, when measuring endothelial permeability in cultured cells, it is difficult to distinguish the contribution of paracellular leakage from true transcytosis [186]. A more serious reason for the poor study of albumin transcytosis by EC is associated with uncertainty about the physiological significance of this process. Mice deficient in caveolin-1 lack caveolae and show decreased endothelial internalization of albumin, but these animals show a compensatory increase in paracellular transport [187]. This model emphasizes the importance of transporting albumin outside the vascular bed but does not provide an answer for the physiological function of transcytosis. Considering the numerous binding sites for FAs [188], it can be assumed that albumin transcytosis is important for the regulated transfer of circulating FAs to various tissues. In this regard, a long-established phenomenon seems to be important: albumin deficiency correlates with increased levels of circulating cholesterol and phospholipids [189,190].

Most published research related to the study of transcytosis was carried out on the lung endothelium, whose cells bind albumin through glycoprotein 60 (gp60) in the caveolae and carry out its transfer with the participation of tyrosine kinases [191,192]. Transcytosis of albumin in the lungs is stimulated by thrombin and this process is associated with an increase in the activity of acidic sphingomyelinase, which promotes the synthesis of ceramide and caveolin-1 and their recruitment to membrane lipid rafts; stimulation of albumin transcytosis by proinflammatory mediators may contribute to the leakage of alveolar proteins during lung damage [193]. However, in the lungs, another mechanism of transcytosis is also of great importance—pinocytosis-like low-affinity uptake of albumin in the liquid phase [194]. Pinocytosis is a process of nonselective absorption by a cell of the liquid phase of the environment, containing soluble substances encapsulated within small vesicles (endosomes), which merge with lysosomes [195].

For a long time, practically nothing was known about albumin transcytosis in other tissues. Considering the heterogeneity of the vascular endothelium in different organs [196], it could be assumed that there are different mechanisms of albumin transcytosis. This assumption was confirmed in the recent work of Raheel et al.; according to their data, transcytosis of albumin in the skin—unlike in lungs—has saturation kinetics, which indicates a receptor-mediated process [197]. It was found that for albumin transcytosis through the endothelium of the microvessels of the dermis, the necessary and sufficient condition is the expression of the CD36 glycoprotein, which is known as the scavenger receptor class 3B (SR-B3), platelet membrane glycoprotein IV (GPIV), glycoprotein IIIb (GPIIIb), thrombospondin receptor, collagen receptor, FA translocase (FAT), and even as an innate immune receptor [198]. Upon ligand binding, CD36 triggers a signaling cascade that mediates a wide range of pro-inflammatory responses. For example, amyloid-β1–40 (Aβ), interacting with CD36, activates the generation of superoxide anions by NADPH oxidase [199]. In acute lung injury, increased levels of ROS and intracellular calcium play a key role in the dysfunction of the endothelial barrier; the H_2_O_2_-induced increase in [Ca^2+^]_i_ in the EC of the microvessels of the lungs is associated with the activation of TRPV4 (type 4 cationic vanilloid channels with a transient receptor potential), with CD36 playing an important role in H_2_O_2_-mediated lung injury through CD36-dependent attachment of Fyn kinase (from the Src family) to the cell membrane, to facilitate phosphorylation of TRPV4 [200].

Glycoprotein CD36 is expressed not only on the surface of ECs of microvessels, but also platelets, monocytes, smooth muscle cells, cardiomyocytes and a number of other cells—but is absent in the lymphatic vessels of the dermis [201]. Despite its widespread distribution, CD36 has long been a rather mysterious protein. Over time, it was found that CD36 can influence cellular responses by interacting with various ligands, in particular, with thrombospondin-1, oxidized LDL and long-chain FAs [202,203]. Currently, CD36 is considered the main membrane protein involved in lipid homeostasis in the body. CD36 acts in concert with membrane and cytoplasmic proteins that bind FAs. The rate of FA uptake depends on the presence of CD36 on the cell surface, which is regulated by subcellular vesicular recirculation of CD36 from endosomes to the plasma membrane [204]. However, non-esterified FAs cannot circulate freely in plasma as they are bound to albumin [205], which has seven FA binding sites [206]. Interactions with CD36 are associated with the activation of Src-family kinases and mitogen-activated protein kinases, as well as with the participation of Rho-GTPases and NFκB transcription factors [207]. Whether binding and transcytosis of albumin requires these or other signaling pathways remains to be investigated.

Mice selectively deficient in endothelial CD36 had a reduced level of subcutaneous fat, while the level of circulating lipids was comparable to control animals and representatives of both groups had similar body weights [197]. This is consistent with a defect in lipid transport or in metabolism at the endothelium–skin interface. According to this hypothesis, analbuminemic rats show hypercholesterolemia [208], and patients with congenital albumin deficiency show elevated serum cholesterol and phospholipid concentrations, which temporarily return to normal after intravenous albumin infusion [189,190]. Mutant or deficient CD36 have not been previously associated with skin phenotype [209], but this is due to the fact that previous research has studied the effects of CD36 deficiency throughout the body, rather than deletions specific for ECs. After investigation of the CD36 role in the skin, the question arises about the role of CD36 in the endothelium of lung microvessels. Although it does not mediate albumin transcytosis, there is evidence that lung endothelial CD36 is involved in the response to inhaled pollutants [210] or infection [211].

However, CD36 is not an exclusive mediator of the effect of FAs on EC. In the absence or with a lack of albumin, inflammation may arise as a result of their interaction with Toll-like receptors, and not as a result of absorption through CD36 [212]. Thus, palmitate, while it is not bound to albumin, activates inflammatory pathways in the ECs of microvessels, enhances the generation and/or expression of IL-6, IL-8, TLR2 (Toll-like receptor 2) and intercellular adhesion molecules 1 (ICAM-1) in them, disrupts insulin transport and promotes monocyte transmigration. Inhibition of CD36 does not affect palmitate-induced expression of adhesion molecules; at the same time, suppression of signaling through TLR4 to NF-κB reduces palmitate-induced expression of ICAM-1 [212]. Such signaling switching options are important for understanding the importance of albumin in both the prevention and development of cardiovascular diseases.

## 7. Interaction of Glycated Albumin with the Endothelium

In diabetic patients, the level of GA is more than 70 μM; it is one of the initiators of EC apoptosis [213]. The likelihood of atherosclerotic plaque formation in the carotid artery in patients with DM2 correlates with an increased level of GA and a decreased level of endothelial progenitors (CD34+/133+/309+) [214]. Interestingly, with the formation of plaques, the level of GA decreased, as well as the ratio of GA to glycated hemoglobin. This is partly due to the fact that in DM, aortic ECs switch to a biosynthetic phenotype with an increased number of caveolae and increased (by about 20%) transcytosis of glycated albumin. In EC cultures, 25 mM glucose causes an approximately 2.6-fold increase in pSTAT-3 and pERK1 and an approximately 1.8-fold increase in pERK2; exposure to glycated albumin (5 μM) causes an approximately 4.3-fold increase in pERK1/2 compared to 5 mM glucose [215].

GA induces the expression of procoagulant and inflammatory factors by ECs (Figure 5) [216,217]. At the same time, there is evidence that the increased level of von Willebrand factor in diabetics is due not to glycated albumin, but to the effect of mannose-specific lectins on the hydrocarbon determinants of ECs [218]. However, it should be mentioned that glycated albumin was tested at low concentrations in a culture of ECs from the umbilical vein (HUVECs) (25–100 μg/mL), while physiologically much more active lectins (concanavalin A, ConA, and wheat germ agglutinin, WGA) were tested at slightly lower concentrations (4–16 μg/mL).

In DM, many morphofunctional abnormalities of cells are mediated by auto- and paracrine TGFβ, which is induced by high levels of glucose in the environment and by glycosylated proteins. For most cell types, TGFβ is an inducer of apoptosis, which is mediated by the type I TGFβ receptor, Alk5. In contrast, early diabetic microangiopathy is characterized by increased proliferation of ECs. Endothelial cells are unique in that they express one more type I TGFβ receptor, Alk1, as well as the endoglin coreceptor, which increases the ligand’s affinity for Alk1. In differentiated ECs of healthy subjects, Alk1 and endoglin are constitutively expressed. However, incubation of ECs with high glucose content and glycosylated albumin in the medium induces Alk5 expression and increases TGFβ secretion 3-fold, without affecting Alk1 or endoglin levels. The “diabetic” environment accelerates cell proliferation, at least in part due to TGFβ/Alk1-smad1/5 and, probably, with the participation of VEGF, as well as promigratory MMP2 downstream of Alk1. In addition, the activity of caspase-3 is partially increased, which indicates an increase in apoptosis via the TGFβ/Alk5-smad2/3 pathway. These data indicate pleiotropy of TGFβ in ECs, including proliferative effects (via Alk1-smad1/5) and proapoptotic signals (via Alk5-smad2/3) [219]. Among other proinflammatory cytokines, the synthesis of which induces GA, TNFα should be noted [220]. IL-6 expression by ECs is also increased by GA, but this increase can be prevented by angiotensin 1-7, which is a product of ACE-2 activity [221].

GA enhances the expression of RAGEs against a background of shear stress (Figure 5) [222] and increases the expression of adhesion molecules VCAM-1, ICAM-1 and E-selectin [223]. The expression of EC adhesion proteins (in particular, E-selectin, or CD62E) increases in a precise manner when exposed to GA, whereas there is no such effect under the action of a heterogeneous mixture of AGEs, which are formed as a result of non-enzymatic glycation and oxidation of proteins, lipids and nucleic acids [224]. GA from patients with heart failure and high glycation levels also increases the expression of adhesion molecules in HUVECs and enhances adhesion of peripheral blood mononuclear cells to ECs [225]. As is known, mononuclear cells are powerful generators of ROS, but what is especially important is that an increase in the expression of adhesion molecules is mediated by the generation of ROS by NADPH oxidase (NOX) in ECs and a signaling pathway involving kinases PKB-IKK and JNK, as well as transcription factors NF-kappaB and AP-1 (Figure 5). It was found by RT–PCR that the generation of ROS is maximally expressed 4 h after the onset of GA exposure and is accompanied by an increase in the expression of NOX4 and p22phox mRNA [226]. Endothelial NOX2 activation also contributes to glomerular dysfunction in insulin-dependent mice, downregulating the expression of the glomerular glycocalyx (Figure 5), and causing morphofunctional changes in podocytes and mesangial cells; ultimately, this contributes to the development of diabetic nephropathy, one of the signs of which is albuminuria [227]. In addition to NOX4, another source of ROS under the action of GA on EC is uncoupled endothelial nitric oxide synthase (eNOS; Figure 5) [228]. GA can both enhance and weaken the activity of NOS, and both variants of the response were noted during the development of EC apoptosis under the influence of GA [213,229,230]. This is consistent with the notion that oxidative stress plays a key role in endothelial damage in DM, and the degree of albumin glycation affects its intensity through a possible connection between NOX, mitochondria and other sources of ROS [231].

Recently, it has been shown that one of the important mediators of AGE-induced diabetic endothelial dysfunction is peroxidazine (PXDN), a member of the peroxidase family, which catalyzes the conversion of hydrogen peroxide to hypochlorous acid. It is believed that NOX2 is a source of ROS upon exposure to AGEs, while the action of HOCl leads to a weakening of eNOS phosphorylation at Ser1177 and a decrease in NO synthesis (Figure 5) [232]. Intracellular paraoxonase-2 is another participant in the antioxidant defense of ECs and is a target of GA. GA, as well as N-carboxymethyllysine (CML, the most famous representative of AGEs) suppress the expression and activity of PON2 in ECs [233]. In addition, the effect of GA and CML on the endothelium leads to an increase in the level of endoplasmic reticulum stress markers GRP78 and IRE1α, as well as to an increase in the expression of proinflammatory cytokines MCP-1, IL-6, IL-8, and adhesion proteins ICAM1 and VCAM1. At the same time, an increase in PON2 expression leads to a decrease in ROS levels and facilitates endothelial dysfunction caused by AGEs [233].

## 8. Role of Modified Albumin in Pathogenesis of Diseases

Despite the availability of major proteins in general, and albumin in particular, there is insufficient data about mechanisms of albumin modification or the mechanisms of modified albumin itself on cells, tissues and health. Moreover, current data is often controversial. The pathogenic specificity of a particular organ is due to the fact that when the endothelium is damaged, the release of albumin outside the vascular bed becomes uncontrolled, which leads to a change in the biological activity of parenchyma cells. In addition, blood cells can also be a target for AGEs and GA, that, as a rule, aggravate the state of ECs and cause distortion of blood–tissue barriers. Thus, GA increases the thrombogenic potential of platelets due to an increase in the number and sensitivity of receptors [234].

The pathogenic role of AGEs and GA is most studied in obesity, diabetic polyneuropathy and nephropathy [132,235,236]. Coronavirus SARS-CoV-2, which caused the COVID-19 pandemic, has stimulated research on the relationship between susceptibility to infection and the presence of other diseases in anamnesis, including those listed above—one of the pathogenic factors of which is GA.

### 8.1. Obesity

Visceral fat is one of the main sources of pro-inflammatory cytokines due to activated macrophages, which are much more abundant in comparison with subcutaneous fat. This is primarily due to the accumulation of GA and increased expression of RAGEs [235]. There are molecules among cytokines that interact with RAGEs (S100β and HMGB-1), and also molecules of which levels increase upon activation by RAGEs (MCP-1, IL-6, TNFα, TGF-βetc.). Chronic inflammation is maintained by constant recruitment of macrophages through RAGE-dependent expression of MCP-1 and further RAGE activation; this self-amplification loop is called the RAGE/MCP-1 axis [237]. Increases in AGE levels, mainly in the form of GA, feeds either oxidative stress or the AGE/RAGE axis, which in turn can potentiate inflammation in already inflamed tissue, thereby accelerating the progression of obesity. Moreover, adipocyte adaptive functions are disrupted by other receptors of AGEs: for example, the entrapping and degradation of AGEs by the CD36 scavenger leads to a decrease in the generation of leptin by adipocytes, which may contribute to the progression of obesity [238].

### 8.2. Diabetic Polyneuropathy

Diabetic polyneuropathy (DPN) processes have been found in all patients with type 1 DM (DM1) within 15 years of diagnosis and in 30% of patients with DM2 within 25 years [239]. A characteristic feature of DPN is EC hyperplasia and thickening of the basement membrane of endoneural capillaries and many other microvessels. The mechanism of this phenomenon remains unclear—to what extent is it related to the polyol pathway of glucose metabolism in ECs, and to what extent is it related to the effect of glycated albumin on RAGEs? One of the molecular pathogenetic factors of DPN is a decrease in the synthesis of claudin-5, the most important component of tight junctions, by ECs of endoneural capillaries as a result of the action of AGEs on EC receptors. This leads to disruption of the barrier function of the endothelium and edema of the nerve fiber. AGEs reduce the amount of claudin-5 indirectly by increasing the autocrine secretion of endothelial growth factor (VEGF) by ECs that form the blood–nerve barrier (BNB) [236]. Moreover, pericytes play an important role in the formation and maintenance of the BNB basement membrane; they produce fibronectin and type IV collagen, as well as a tissue inhibitor of metalloprotease-1 (TIMP-1), which prevents degradation of the basement membrane. It turned out that during hyperglycemia, AGEs accumulate in pericytes, increasing the autocrine secretion of VEGF and TGF-β. Signaling from VEGF and TGF-β potentiates the production of fibronectin and type IV collagen, which leads to a thickening of the basement membrane [236].

It’s interesting to note that treatment of hyperglycemia reduces the frequency of DPN by 60–70% in patients with DM1 [240], and by only 5–7% in patients with DM2 [241]. Moreover, DPN progresses in at least at 40% of patients with DM2 even with controlled levels of glucose [242]. There is not enough data about the role of modified (glycated and/or oxidized) albumin in DPN pathogenesis to seriously think about the prospects for developing effective therapies.

### 8.3. Diabetic Nephropathy

Each kidney contains approximately one million nephrons, with each nephron consisting of a glomerulus and tubules. Glomeruli consist of four types of cells: parietal epithelium, glomerular endothelium (glomerular endothelium cells, GECs), podocytes (visceral epithelial cells), and mesangial cells. Endothelium and podocytes have a common extracellular matrix—the glomerular basement membrane (GBM). GECs and intercellular clefts are covered by endothelial glycocalyx; these are the appendices with slotted diaphragms which surround the external part of the capillaries on podocytes. GECs with an endothelial glycocalix, GBM and podocytes function as the filter barrier of the kidney, also known as the glomerular filtration barrier. When DM develops, the signal exchange between cells is hindered as a result, and during that process, the primary increase in the synthesis of vascular endothelial growth factor A (VEGFA) by podocytes observed in the early stages, is replaced by a decrease in VEGF synthesis during the progression of the disease. There is also a loss of interaction between angiopoietin-1 (Angpt1) and the tyrosine protein kinase receptor (Tie2), and the production of activated protein C (APC) in the glomeruli is reduced due to suppression of thrombomodulin expression.

A decrease in the functional activity of APC affects the porosity of the glomerular capillary wall and potentiates apoptosis of glomerular ECs and podocytes. Metabolic changes associated with DM, along with activation of the renin–angiotensin–aldosterone system (RAAS), cause the generation of ROS and AFAs (nitric oxide, nitrogen dioxide, and peroxynitrite) in GECs. Endothelin-1 (Edn1) activity in DM increases oxidative stress, which causes endothelial nitric oxide (NO) depletion and degradation of endothelial glycocalyx [243].

However, dysfunction of the glomerular endothelium is characterized not only by damage to the endothelial glycocalyx and oxidative stress in ECs, but also by the endothelial–mesenchymal transition (EndMT) [244]. EndMT is a process by which ECs lose their endothelial phenotype (e.g., decreased expression of EC markers CD31 and CD144) and endothelial-specific functional characteristics (atrombogenicity, barrier functions). The decrease in endothelial marker expression comes with an increase in the expression of mesenchymal markers, such as α-smooth muscle actin (αSMA) and fibroblast specific protein 1 (FSP-1); in addition, the synthesis of extracellular matrix proteins (ECM) increases [245]. EndMT promotes the development of fibrosis and is observed in diseases of a wide variety of organs, including cancer [246,247,248]. There is EndMT in patients with diabetic nephropathy that is evidenced by coexpression of endothelial and mesenchymal markers [249]. Hyperglycemia, AGEs, GA, hypoxia and a number of other factors cause dysfunction of the glomerular endothelium, characterized by damage to the endothelial glycocalyx, oxidative stress, an inflammatory phenotype of ECs and EndMT; this leads to proteinuria, damage to or loss of podocytes, activation of mesangial cells, and, ultimately, glomerulosclerosis. An additional pathway of kidney damage in DM is the transdifferentiation of renal tubular cells into myofibloblasts. This occurs when RAGEs are activated, which induces the expression of TGF-β and other cytokines that mediate this transdifferentiation [250]. NF-κB activation, along with RAGE amplification and cytokine expression, causes the activation of the ZEB2 gene and production of a transcription factor that leads to loss of podocyte adhesion, epithelial–mesenchymal transition, detachment of the basement membrane and loss of podocytes in the glomeruli [251]. Activation of protein kinase C (PKC), TGF-beta and gene expression in mesangial cells, along with a similar effect on ECs of the glomerular apparatus, is also a cause of diabetic nephropathy, especially in DM1 [252,253].

It is important to note that EndMT is not irreversible, as the possibility of reverse reprogramming of transformed ECs has been shown [245]; also, EndMT is under autophagic control [254,255].

### 8.4. COVID-19

In addition to numerous observations that evidence modifications and decreases in albumin levels in various diseases accompanied by inflammatory processes, including in many patients with coronavirus infection [256], it has been suggested that the cytokine storm observed in patients with COVID-19 is caused by increased levels of oxidized albumin [257]. At the same time, RAGEs play an important role in the pathogenesis of lung diseases such as fibrosis, pneumonia, and acute respiratory distress syndrome (ARDS). Overexpression/hyperactivation of RAGEs increases the negative effects of renin–angiotensin system (RAS) mediators of chronic diseases that are the main risk factors for coronavirus infection: DM, and kidney and cardiovascular diseases [258]. SARS-CoV-2 enters a cell after recognition and binding of the coronavirus spike protein by angiotensin-converting enzyme type 2 (ACE2), which leads to suppression of ACE2 regulation and increases in angiotensin II (AngII). Infected cells undergo pyroptosis and unleash DAMP (damage-associated molecular patterns), including HMGB1.

DAMP, cytokines and RAGE ligands lead to increased expression of RAGEs by NF-κB. Under these conditions, RAGEs induce further suppression of ACE2 and increases the expression of AT1R (type 1 AngII receptor) and transactivate with AT1R, amplifying the pathogenetic effects of the ACE/AngII/AT1R axis.

Concurrently, AngII/AT1R induces the activation of NF-κB and frees the ligands of RAGEs, so a vicious circle forms. The RAGE interacts with effectors of the renin–angiotensin system (RAS), promoting the progression of a “cytokine storm” in macrophages and splenocytes; causing endothelial dysfunction on account of increased capillary permeability and release of damage-associated molecular pattern components (DAMP); increasing the production of ROS and the formation of atherosclerotic plaques; increasing the risk of thrombosis by inducing the formation and release of extracellular traps by neutrophils (NET), with further thrombocyte aggregation; and causing muscle wasting, stimulating apoptosis, increasing protein degradation and decreasing protein synthesis. The combined effects of RAGEs and AT1R occur in the vessels and parenchyma of the lungs, brain, heart and kidneys, and in the cells of the immune system, causing irreversible damage to many organs [259]. The question of what is still primary (cause) and what is secondary (consequence), in this case, not only has the right to be asked, but deserves serious attention on the part of scientists and physicians.

### 8.5. The Role of Albumin in Epileptogenesis

Recent research findings strongly suggest that AGEs are the main contributors to brain microvascular damage and disruption of the blood–brain barrier (BBB) [260]. Interactions between the BBB, cerebral blood vessels, neurons, astrocytes, microglia and pericytes form a dynamic functional neurovascular unit. Damage to the cerebral cortex as a result of trauma, intoxication, ischemia or infection can lead to the progression of post-traumatic epilepsy (PTE), one of the most common neurological disorders—the pathogenesis of which is closely related to distortion of BBB integrity [261,262]. This, in turn, leads to the leaking of plasma components into the brain parenchyma and increased excitability of neurons—firstly as a result of a primary increase in K^+^ and glutamate levels; then, as conjugated mechanisms are launched, extravasated albumin is absorbed by astrocytes through TGF-βR and leads to Smad2-mediated suppression of potassium channel Kir4.1, whereas astrocytic TNF-α initiates a decrease in expression of the glutamate transporter EAAT-2. Both mechanisms aggravate primary neuronal hyperactivity due to disrupted buffering of K^+^ and glutamate by astrocytes, resulting in extracellular potassium accumulation, relief of NMDA-mediated hyperexcitability, and finally, epileptiform activity [263,264,265]. It should be noted that the same signal pathway is activated in aging patients with BBB dysfunction [266]. Currently, there are no tools to identify patients at risk of progression in PTE, or to prevent this progression. Seizures that may occur months or years after stroke, do not react to anticonvulsants in more than a third of patients, and are often associated with significant neuropsychiatric illness [267]. The increased concentration of albumin causes increases in [Ca^2+^] with the help of inositol-1,4,5-thriphosphate (IP3). Besides that, albumin induces the synthesis of DNA. These processes are partially blocked by heparin and TGF-β antagonists [268]. Thus, the use of SJN2511, which is a specific inhibitor of the ALK5/TGF-β pathway, prevents excitatory synaptogenesis and albumin-induced epilepsy [267]. Use of non-specific drugs such as losartan (antagonist AT1) also prevents violation of the BBB and the development of epileptogenesis [262]. However, the issue of differences in the affinity of unmodified and modified albumin for cytokine receptors of the brain parenchyma remains unexplored. The dosing of specific and nonspecific drugs for PTE therapy, taking into account the pathogenetic role of albumin, is one of the most pressing problems of modern pharmacology.

## 9. Integrative Properties of Albumin in Diagnostics and Therapy

Albumin levels in plasma or serum have been a classic marker of nutritional status for many years, especially for protein foods. A level of less than 35 g/L is defined as hypoalbuminemia. Recently, low albumin levels are increasingly considered a risk factor and predictor of morbidity/mortality, regardless of gender, age, comorbidities and all kinds of polymorphisms [269,270,271]. Several papers report on the role of albumin levels in plasma and various human diseases such as infections [272,273], cancer [274,275] and even depression in HIV-infected patients [276]. The level of albumin in plasma and urine reflects the protein-synthesizing function of the liver (hence there it has a role as a negative acute-phase protein [277]) and the functional state of the vascular endothelium, which determines the integrity of blood–tissue barriers. Therefore, from the point of view of diagnostics, the level of albumin characterizes not just the level of one protein, but its integrative characteristics assess the state of the whole organism. The relationship between the integrity of the endothelium and the level of albumin in urine is the most studied phenomenon in medical practice, indicating primarily kidney pathology, but also the state of other components of the blood and cardiovascular system [278,279,280]. Among these components, the vascular endothelium of the brain is of the greatest interest, since on the other side of the barrier there is the brain parenchyma, the “holy of holies” of the whole organism. The functional units of the parenchyma turned out to be extremely vulnerable to the action of albumin due to the presence of a special receptor on astrocytes, which until recently was considered specific to TGF-β, one of the minor cytokine proteins that regulate cell differentiation and apoptosis [267,268]. In the period at the beginning of or even in the heyday of studies of cytokine regulation, it was difficult to even imagine a common receptor for proteins, the difference in concentration of which in the blood serum is more than 9 orders of magnitude. The difference in affinity for the receptor is compensated for by the potential superiority in the number of albumin molecules that can access astrocyte receptors when the BBB integrity is disrupted. In this regard, the cytotoxic characteristics of redox-modified and glycated albumin are of great interest not only in relation to ECs of blood vessels, but also in relation to cytokine receptors of astrocytes.

A large-scale search for new diagnostic indicators using modern metabolomics technologies made it possible to pick out only four simple indicators of blood plasma, that allow accurate assessment of the state of human health and prediction of the likelihood of death for patients, regardless of their age, gender and the nature of existing or previous diseases. Among these four indicators, the level of albumin was in second place after orosomucoid (alpha-1-acid glycoprotein) in terms of the degree of contribution to the integral assessment and prognosis [270]. A similar result was obtained in the study of patients in intensive care units (ICUs). The use of a simple ratio of positive (C-reactive protein) and negative (albumin) acute phase proteins can significantly increase accuracy in assessing the risk of death [281]. A structurally similar ratio with markers of muscle fiber damage (creatine kinase or myoglobin) in the numerator significantly increases the correlation of biochemical indicators with functional and physiological (instrumental) indicators [282]. The urine albumin/creatinine ratio is one of the most sensitive indicators of glomerular renal dysfunction and hypertension in patients with high-risk neuroblastoma treated with myeloablative regimens [283]. During the coronavirus pandemic, it was found that there is an inverse correlation between mortality from COVID-19 and the concentration of albumin in the blood [284]. The study authors suggested that this association may be related to the anticoagulant and antioxidant properties of albumin.

Competent application of regression analysis methods makes it possible to increase the sensitivity and specificity of diagnosing diabetic complications through the use of “internal” indicators of albumin, such as the ratio of its reduced and oxidized forms [285]. The ratio of oxidized albumin to total albumin can increase in liver disease, DM, and cardiovascular disease—leading to bacterial or viral infections. During the pandemic, it was found that levels of oxidized albumin in the blood of patients with COVID-19 can be a positive predictor of mortality due to the induction of a cytokine storm [257].

However, albumin can serve not only as a biomarker of the severity of various pathologies, but also as a means of therapy. Due to its critical physiological role, HSA is one of the most sought after biopharmaceuticals. Currently, the annual demand for HSA worldwide is estimated at about 500 tons [286]. Injections of 5% albumin solution (isooncotic solution) are prescribed if necessary to increase intravascular volume, injections of 20–25% albumin (hyperoncotic solution) are prescribed to restore COP and maintain fluid balance between intravascular and extravascular compartments [287]. Clinical indications for the use of a 4–5% albumin solution are hypovolemic shock, acute liver failure, and cardiopulmonary bypass [288]. Indications for use of a 20–25% solution are hypoalbuminaemia, sepsis or septic shock, and cirrhosis with ascites [289]. At the same time, one should keep in mind that routine correction of hypoalbuminaemia in critically ill patients is not advised, and its use in sepsis and septic shock remains debated. Continuous infusion of a solution with 4% albumin in ICU patients reduces the risk of nosocomial infections [290]. According to the data obtained, albumin reduces the oxidized form of vasostatin-1 and thereby restores its antimicrobial properties. Albumin can be used to deliver organosulfur compounds to melanoma cells in order to inhibit melanin synthesis [291]. The ability of albumin to bind water can be used in the treatment of OP poisoning. A decrease in glycocalyx density leads to a decrease in COP and hypovolemia, so that appropriate compensation could become one of the therapeutic factors in acute OP poisoning to reduce the risk of death and prevent delayed pathology. Indeed, cases of successful use of fresh frozen plasma in the treatment of the so-called “intermediate syndrome”, one of the possible consequences of OP poisoning, have been described [292].

At the same time, the introduction of albumin can be dangerous in some situations. Cardiac and renal failure, acute or chronic pancreatitis, pulmonary edema, and severe anemia are the conditions in which the use of albumin is contraindicated due to the risk of acute circulatory overload. Also, the use of albumin is not recommended for conditions such as ascites responsive to diuretics, non-hemorrhagic shock, hypoalbuminemia without edema or acute hypotension, malnutrition, open wounds, acute normovolemic hemodilution in surgery, and protein loss due to enteropathy or malabsorption. In addition, the use of albumin can be life-threatening for patients with cerebral ischemia and traumatic brain injury [287,293].

From a physiological point of view, the simplest and most natural way of influencing the level and properties of albumin is so-called “functional food”. Functional food is the use of such products and methods of processing in foods that would maximally preserve the quantity and useful qualities of the nutraceuticals they contain. These nutraceuticals, in turn, would have a positive effect on both the intestinal microflora and functional state of the liver, where partial metabolism of nutraceuticals and albumin synthesis occurs. Moreover, they would have an effect on the state of albumin in the blood plasma, because many polyphenols and other nutraceuticals are bound and transported in the systemic circulation, mainly by albumin [294], changing its conformation and competing with other ligands, including FAs and AGEs.

The relationship between GA (and other food-borne AGEs) and gut microbiota has recently become the subject of research [295]. The need for the development of standardized methods for determining the rate of consumption of AGEs is rightly indicated. Numerous studies have shown that polyphenols largely determine the number, composition and condition of intestinal bacteria, which in turn modulate neurological diseases [296]. Polyphenols have not only antioxidant properties, but also the ability to protect proteins from glycation (antiglycation ability). The consumption of polyphenols with food causes an increase in the peripheral blood concentration of phenolic acids, the most abundant of which is 3-hydroxyphenylacetic acid (up to 338 μM). The concentrations of other phenolic acids are in the range of 13 nM to 200 μM. An in vitro experiment showed that pre-incubation of BSA with various phenolic acids and subsequent glycoxidation of albumin (glucose 5–10 mM in combination with 10 nM H_2_O_2_) significantly reduces the concentration of fructosamine [297]. Chrysin and luteolin, which are structurally related flavone aglycones, are common polyphenols that can reduce the degree of albumin glycation. They are found in broccoli, chili peppers, celery, rosemary, and honey [154]. Luteolin and chrysin display anticarcinogenic and cardioprotective properties, in particular, due to their ability to neutralize ROS, to suppress the expression of cyclo-oxygenase-2 and the formation of prostaglandin E2 [298]. The polyphenol-rich extract of the medicinal plant *Doratoxylon apetalum* has proved to be an effective antioxidant for protecting ECs by decreasing the level of hydrogen peroxide and superoxide in them [297]. In addition to polyphenols, garlic extract also inhibits the formation of AGEs, including glycated albumin [299].

Aging-related gluco-oxidative stress accompanied by AGE formation is thought to generate neoepitopes on blood proteins, thus promoting the production of autoantibodies in the elderly, especially in smokers. The use of natural products with antioxidant nutraceuticals reduces the manifestation of age-related pathophysiological changes. The mechanism of the protective action of polyphenols is not fully understood, but it is assumed that polyphenols interact non-covalently with the aromatic amino acid residues of albumin. This hydrophobic interaction promotes the remodeling of mature AGE-modified amyloid fibrils and transforms the secondary structure into a helical or disordered helical conformation [166]. Chrysin and luteolin inhibit the formation of albumin fibrils. Molecular docking showed that both flavonoids non-covalently interact with various amino acid residues of the IIA subdomain, including lysines and arginines, which are prone to glycation. The flavonoids additionally stabilize the HSA structure, which explains the mechanism of their action as antiglycating and antifibrillating agents [154]. It is assumed that polyphenolic compounds have pleiotropic effects and prevent glycation at different levels, through the regulation of glucose metabolism, metal chelation, trapping of intermediate dicarbonyl compounds, influence on the insulin resistance of cells, and finally, through the activation of the signaling pathway of the insulin-like growth factor receptor [300].

## 10. Conclusions

Thus, the characteristics of albumin revealed in recent years indicate that this major blood plasma protein, which until recently was assigned the modest role of an osmotically active component, is in fact a molecular “core” and a link between various tissues and organs, indicating the health of the whole organism, and in many respects, determining this health. Modern diagnostics, the pathogenesis of various diseases and the development of therapeutic agents are currently unthinkable without a comprehensive consideration of the physicochemical, evolutionary–genetic, and physiological–biochemical characteristics of albumin.

## Figures and Tables

**Figure 1 ijms-22-10318-f001:**
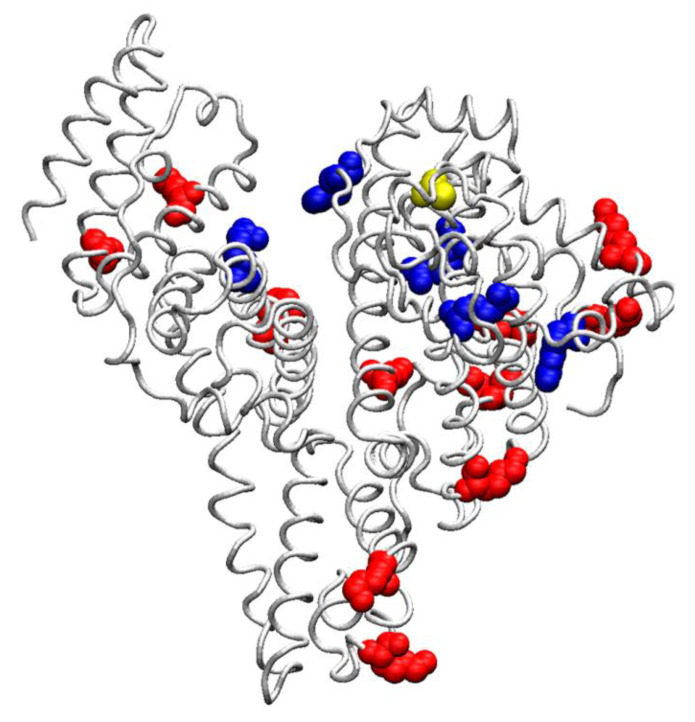
The main sites of HSA modification. The sites of in vivo glycation are shown in red (lysines) and blue (arginines). The site of redox modification (Cys34) is shown in yellow. To create the figure, a crystal structure of HSA from the PDB database (code 3JQZ [24]) was used.

**Figure 2 ijms-22-10318-f002:**
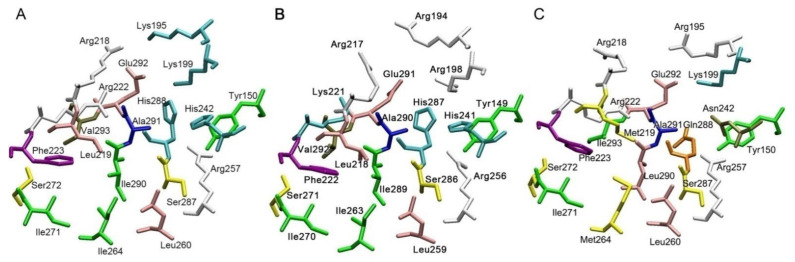
Structures of Sudlow site I of HSA (**A**), BSA (**B**) and RSA (**C**).

**Figure 3 ijms-22-10318-f003:**
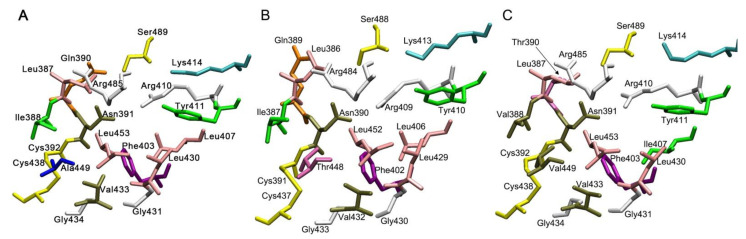
Structures of Sudlow site II of of HSA (**A**), BSA (**B**) and RSA (**C**).

**Figure 4 ijms-22-10318-f004:**
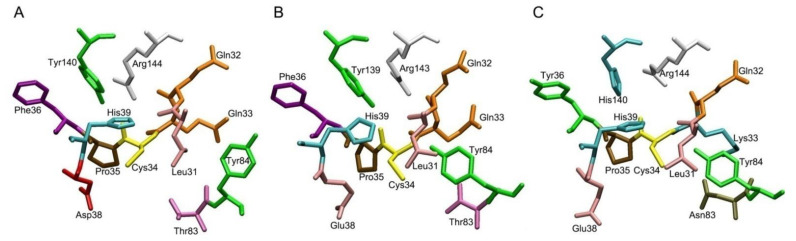
Structures of the redox sites of HSA (**A**), BSA (**B**) and RSA (**C**).

**Figure 5 ijms-22-10318-f005:**
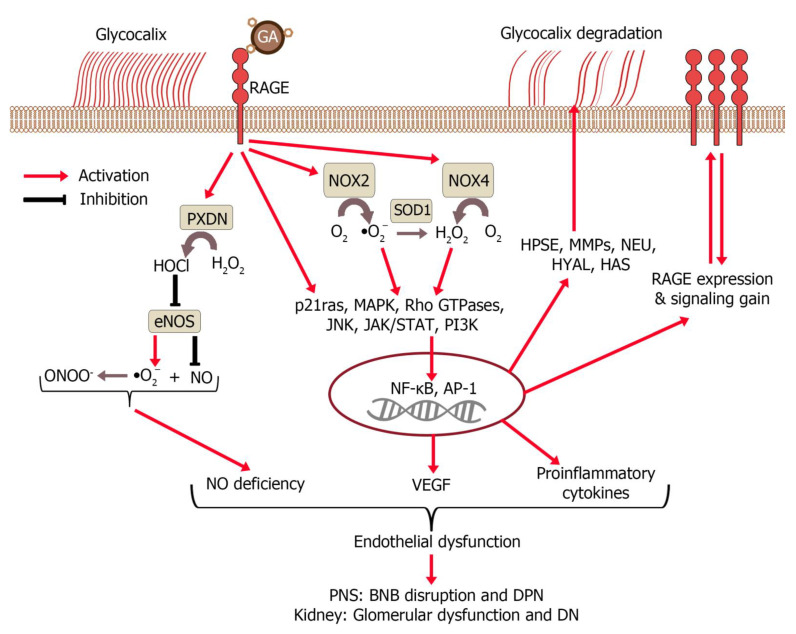
Interaction of glycated albumin with endothelial cells (EC). AP-1, activator protein 1; BNB, blood–nerve barrier; DN, diabetic nephropathy; DPN, diabetic peripheral neuropathy; eNOS, endothelial nitric oxide synthase; GA, glycated albumin; HAS, hyaluronic acid synthase; HPSE, heparanase; HYAL, hyaluronidases; JAK/STAT, Janus kinase signal transducer and activator of transcription; JNK, c-Jun N-terminal kinase; MAPK, mitogen-activated protein kinase; MMPs, matrix metalloproteinase; NEU, neu raminidase; NF-κB, nuclear factor kappa-light-chain-enhancer of activated B cells; NOX, NADPH oxidase; p21ras, products of the ras (rat sarcoma virus) gene family; Pho GTPase, Rho family of GTPases; PI3K, phosphoinositide 3-kinase; PNS, peripheral nervous system; PXDN, Peroxidasin; RAGE, receptor for advanced glycation endproducts; SOD, Superoxide dismutase; VEGF, vascular endothelial growth factor.

**Figure 6 ijms-22-10318-f006:**
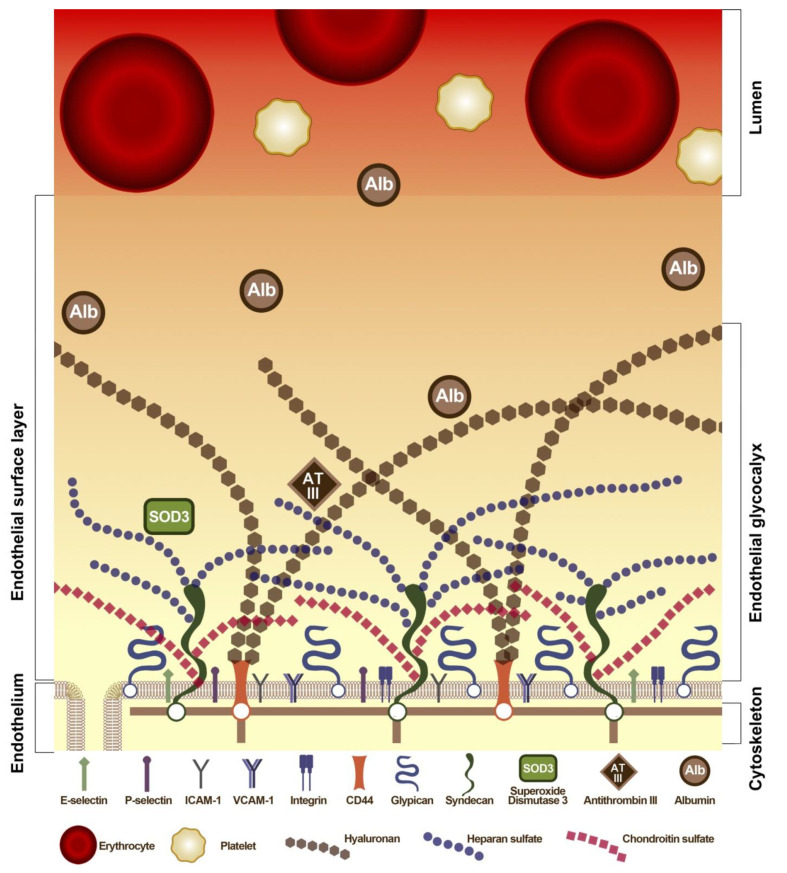
Albumin and endothelial glycocalyx layer (EGL). The EGL serves as the “exclusion zone” or “gap” between endothelium and blood cells. The layer also acts as a filter, limiting the passage of molecules larger than 70 kDa. Albumin, with a molecular weight of about 66 kDa, can firmly bind to the EGL and penetrate into the interstitium by transcytosis.

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
