# Peer review of "Serum Albumin in Health and Disease: Esterase, Antioxidant, Transporting and Signaling Properties"

_ijms, 2021, doi:10.3390/ijms221910318_

Round 1
Reviewer 1 Report
The review by Belinskaia et al. describes in depth several aspects of the physiological role of albumin. I find the work very interesting and important for the field, it provides a very thorough review on various studies on albumin. However, I believe in some aspects, especially structural, the work could be improved. In Chapter 3 the authors compare human, bovine, and rat serum albumins. The choice of the species in this comparison is reasonable since albumins are important from the medical and scientific point of view. Additionally, authors are specialized in the studies on rat serum albumin. However, I do not understand why the structural data of other albumins were omitted. In the PDB there are structures of seven different albumin species. Of course, most of the structural data on the albumin binding features is obtained from the studies on human serum albumin, but in my opinion other species are worth to be mentioned as well. In the recent structural literature, we can find a few interesting structural comparisons of the binding characteristics of different albumin species, like bovine, equine, or rabbit.
Other minor issues:
On the page 2 I believe there is misspelled abbreviation FcR instead of FcRn
On the page 3 the authors cite the paper by Bujacz from 2012 as a reference to the structure of BSA. It should also be the work by Majorek et al., (2012) Mol Immunol 52: 174-182. Both groups have determined the structure of BSA (and other albumins) independently around the same time.
Figures 2-4 would be clearer without displayed hydrogen atoms.
Author Response
We are grateful to Reviewer-1 for the valuable comments on the manuscript. We have substantially modified the manuscript to address the points raised. The changes are detailed below and highlighted in the revised and improved manuscript.
In Chapter 3 the authors compare human, bovine, and rat serum albumins. The choice of the species in this comparison is reasonable since albumins are important from the medical and scientific point of view. Additionally, authors are specialized in the studies on rat serum albumin. However, I do not understand why the structural data of other albumins were omitted. In the PDB there are structures of seven different albumin species. Of course, most of the structural data on the albumin binding features is obtained from the studies on human serum albumin, but in my opinion other species are worth to be mentioned as well. In the recent structural literature, we can find a few interesting structural comparisons of the binding characteristics of different albumin species, like bovine, equine, or rabbit.
The following abstract was added at the very end of Chapter 2 (former Chapter 3):
In addition to HSA and BSA, three-dimensional structures of leporine (LSA) and equine (ESA) [Bujacz, 2012], caprine (CapSA) and ovine (OSA) [Bujacz et al., 2017] serum albumins, as well as structures of recombinant canine (rCanSA) [Yamada et al., 2016] and feline albumins (rFSA) [Yokomaku et al., 2018] have been obtained so far. It is worth to mention the main differences found when comparing these albumins.
The ligand-binding pockets in BSA, ESA and LSA [Bujacz, 2012] revealed different amino-acid compositions and conformations in comparison to HSA in some cases; however, much more significant differences were observed on the surface of the molecules. The hydrophobic residues located at the bottom of Sudlow sites are more highly conserved than the polar residues located at their entrances. The changes in protein sequence adjust the shape and charge distribution of the pockets and modulate the affinity of particular albumins for selective ligands [Bujacz, 2012].
A comparison of OSA and CapSA with the closely related bovine serum albumin (BSA) [Bujacz et al., 2017] revealed that, despite 98% sequence similarity, OSA binds only two molecules of 3,5-diiodosalicylic acid (DIS), whereas CapSA binds six molecules of this ligand. In BSA, DIS molecules are bound at four positions. Two additional binding positions for DIS in BSA are located in fatty-acid binding site 1 in the IB domain and in Sudlow site II between domains II and III. In the CapSA, two additional locations (absent in BSA) for binding of DIS are observed. They are localized in niches on the surfaces of domains I and III. Additionally, analysis of the electrostatic surface potential of serum albumins revealed some differences in the distribution of positive and negative charges, which means that interactions with other proteins found in nature, especially with antibodies, may be different for albumins from various species [Bujacz et al., 2017].
The environment around the Cys34 in rCanSA and rFSA is more polar and flexible compared to HSA, which explains why free sulfhydryl group ratio of the Cys34 is lower in cat and dog albumins compared to HSA [Yamada et al., 2016; Yokomaku et al., 2018]. Interestingly, warfarin (WRF) and phenylbutazone (PBZ) (ligands of Sudlow site I) cannot bind to canine albumin, though superposition of rCanSA and HSA shows that the architectures of their Sudlow sites are identical and the side chains of Tyr150, Lys199, and Arg222 are located at the same positions [Yamada et al., 2016]. It might be that surface charge distribution is responsible for these binding features.
Bujacz, A. Structures of bovine, equine and leporine serum albumin. Acta Crystallogr. Sect. D Biol. Crystallogr. 2012, 68, 1278–1289.
Bujacz, A.; Talaj, J.A.; Zielinski, K.; Pietrzyk, A.J.; Neumann, P. Crystal structures of serum albumins from domesticated ruminants and their complexes with 3,5-diiodosalicylic acid. Acta Crystallogr. Sect. D Struct. Biol. 2017, 73, 896–909.
Yamada, K.; Yokomaku, K.; Kureishi, M.; Akiyama, M.; Kihira, K.; Komatsu, T. Artificial Blood for Dogs. Sci. Rep. 2016, 6, 36782.
Yokomaku, K.; Akiyama, M.; Morita, Y.; Kihira, K.; Komatsu, T. Core–shell protein clusters comprising haemoglobin and recombinant feline serum albumin as an artificial O2 carrier for cats. J. Mater. Chem. B 2018, 6, 2417–2425.
On the page 2 I believe there is misspelled abbreviation FcR instead of FcRn
As we have found in Pubmed, FcRn is more common abbreviation for the neonatal Fc receptor. We would like to ask to keep «FcRn» abbreviation, if possible.
On the page 3 the authors cite the paper by Bujacz from 2012 as a reference to the structure of BSA. It should also be the work by Majorek et al., (2012) Mol Immunol 52: 174-182. Both groups have determined the structure of BSA (and other albumins) independently around the same time.
Corrected
Figures 2-4 would be clearer without displayed hydrogen atoms.
New pictures have been prepared and inserted to the manuscript
Reviewer 2 Report
In this work authors give an excellent overview on the role of serum albumins in living systems considering their decisive role in the transport of any bioactive molecules. In general, the work is interesting, carefully prepared and earns probably high interest justifying publication in the International Journal of Molecular Sciences. The references are upto date, the structure of the manuscript is appropriate, the data collected support the conclusions. I suggest publication with minor revision considering the following remarks:
Authors mention the role of albumins in the transport of “electrically neutral molecules”. I think interactions at least several charged toxin molecules with albumins must to be mentioned and discussed here. Enhanced adsorption of neutral species also must to be discussed.
Competitive binding of e.g. toxins with flavonoids onto the serum albumins also must to be mentioned and discussed here.
Minor Remarks
Abstract: The transport not limited for the neutral species, I suggest to change this in the abstract.
Author Response
We are grateful to Reviewer-2 for the valuable comments on the manuscript. All of the points were thoroughly considered, and a new section (2.1), several related sentences to existing sections and corresponding references were introduced to the manuscript. The abstract was corrected.
Reviewer 3 Report
The review by Belinskaia and coworkers aims at revising the role of human serum albumin in health and disease. The review highlights some interesting points, although the overall content is quite confused. Indeed, it is not clear the rationale in selecting the topics treated in this review. The Authors jump from evolutionary, to enzymatic, to structural, to pathological topics, without entering really in detail and without citing properly the literature. For sure dealing with this topic is difficult, considering the very high number of papers related to the multiple structural, functional, and therapeutical aspects of HSA, and for this reason the aim of this review must be better focused. The text is hard to be followed in the actual form, as a very high number of topics are treated but none of them are really detailed. For this reason, also considering the tile of the review, I would suggest to better focus the attention on a more limited number of topics and analyze them more in detail.
- I suggest dividing the first, fourth, and fifth paragraphs i subparagraphs. Indeed, in the actual form, these paragraphs are full of information that are very different each other and therefore require to be ordered.
- The title of paragraph 5 “Glycated albumin: biomarker and pathogenetic factor of diabetes mellitus. Advanced glycation end-products and their receptors” seems two titles. This title, as well as the very heterogenous contents of the paragraph, must be revised.
- I suggest making some order in the whole text, which is confused in its subdivision. It is necessary to reorder the contents in a more linear way, avoiding a “ping-pong” discussion that renders very difficult to follow the logic and the aims of the work.
- Attention to the acronyms: for example, the acronyms FA and HSA have been introduced in the first two pages and must be used throughout the text.
- Although the Authors explain the choice of speaking about the “Comparative characteristics of human, bovine and rat albumin” (paragraph 3), this topic is not linked to the rest of the review. This paragraph seems to interrupt the logic of the text.
- Paragraph 8: the Authors have discussed the role of modified albumin in pathogenesis of diseases. Why did they discuss only the role of modified albumin? For example, several papers report the role of unmodified albumin and of its circulating levels in plasma in human diseases (e.g., cancer, infections). The rationale behind this choice is not clear.
- Figure 6: the meaning of this figure is not clear. The caption of the figure indicates “Albumin and endothelial glycocalyx” but what should the reader comprehend from this figure? This figure illustrates the structure of the EGL, without providing any specific information on albumin.
- The conclusions are not real conclusion, but a further paragraph. Indeed, the tile of this paragraph is not just “Conclusions” but “Conclusions: integrative properties of albumin in diagnostics and therapy.” This must be avoided, as conclusion should highlight the most important “take home message”.
Author Response
We are grateful to Reviewer-3 for the valuable comments on the manuscript. We have substantially modified the manuscript to address the points raised. The changes are detailed below and highlighted in the revised and improved manuscript.
The review highlights some interesting points, although the overall content is quite confused. Indeed, it is not clear the rationale in selecting the topics treated in this review. The Authors jump from evolutionary, to enzymatic, to structural, to pathological topics, without entering really in detail and without citing properly the literature. For sure dealing with this topic is difficult, considering the very high number of papers related to the multiple structural, functional, and therapeutical aspects of HSA, and for this reason the aim of this review must be better focused. The text is hard to be followed in the actual form, as a very high number of topics are treated but none of them are really detailed. For this reason, also considering the tile of the review, I would suggest to better focus the attention on a more limited number of topics and analyze them more in detail.
We have rearranged the text to make the content more logical and divided some chapter to subsections for easier reading. In our further review papers, according to advice of Reviewer-3, we will focus the attention on a more limited number of topics and analyze them more in detail.
I suggest dividing the first, fourth, and fifth paragraphs i subparagraphs. Indeed, in the actual form, these paragraphs are full of information that are very different each other and therefore require to be ordered.
We would like to ask not to divide chapter 1 to subsections, since genetic and evolutionary aspects are tightly connected and it is difficult to separate them. However we have rearranged Сhapter 1 to make it more logical: classification àevolution àgene properties à synthesis à posttranslational modifications
Chapter 4 was divided into two subsections: 4.1. Antioxidant properties of albumin and 4.2. Practical aspects of redox status of albumin. Content of chapter 4 has been rearranged according to these subsections.
Chapter 5 has been divided to subsections: 5.1. The role of AGE and GA in DM pathophysiology; 5.2. GA as a diagnostic tool for DM; 5.3. Species differences in glycation properties of albumin
The title of paragraph 5 “Glycated albumin: biomarker and pathogenetic factor of diabetes mellitus. Advanced glycation end-products and their receptors” seems two titles. This title, as well as the very heterogenous contents of the paragraph, must be revised.
The title of chapter 5 has been shortened: Glycated albumin: biomarker and pathogenetic factor of diabetes mellitus”. The chapter has been divided to subsections 5.1. The role of AGE and GA in DM pathophysiology; 5.2. GA as a diagnostic tool for DM; 5.3. Species differences in glycation properties of albumin
I suggest making some order in the whole text, which is confused in its subdivision. It is necessary to reorder the contents in a more linear way, avoiding a “ping-pong” discussion that renders very difficult to follow the logic and the aims of the work.
We have changed the order of chapters 2 and 3 (“Comparative characteristics of human, bovine and rat albumin structure” is now Сhapter 2, and “Enzymatic activity of albumin” is Сhapter 3) to make content more logical: evolutionary and genetics aspects (chapter 1) à structural features (chapter 2) à enzymatic activity (chapter 3) etc. We also have changed the name of Chapter 2 (former Chapter 3) to reflect the content of this paragraph more precisely. The new name is: “Transporting function and structural characteristics of albumins of different species”. Additionally, we have changed the name of Chapter 1; the new one is “Introduction: evolutionary and genetic features of albumin”. The paragraph devoted to comparative analysis of primary sequence of HSA, BSA and RSA has been moved from Chapter 1 to Chapter 2. The paragraph devoted to domain arrangement of albumin has been from Chapter 1 to Chapter 2, too. The paragraph devoted to interaction of albumin with endogenous and exogenous ligands has been transferred from Chapter 1 to Chapter 2. The paragraph devoted to mercaptalbumin and non-mercaptalbumin has been transferred from Chapter 1 to Chapter 4. Chapters 4 and 5 have been divided into subsections.
Attention to the acronyms: for example, the acronyms FA and HSA have been introduced in the first two pages and must be used throughout the text.
Corrected (including other acronyms, like BSA (bovine serum albumin), EC (endothelial cell), DM (diabetes mellitus), etc.)
Although the Authors explain the choice of speaking about the “Comparative characteristics of human, bovine and rat albumin” (paragraph 3), this topic is not linked to the rest of the review. This paragraph seems to interrupt the logic of the text.
As it was mentioned above, we have changed the order of Chapters 2 and 3 (“Comparative characteristics of human, bovine and rat albumin structure” is now Chapter 2, and “Enzymatic activity of albumin” is Chapter 3) to make content more logical: evolutionary and genetics aspects (Chapter 1) à structural features (Chapter 2) à enzymatic activity (Chapter 3) etc. We also have changed the name of Chapter 2 (former Chapter 3) to reflect the content of this paragraph more precisely. The new name is: “Transporting function and structural characteristics of albumins of different species”. Additionally, we have changed the name of Chapter 1; the new one is “Introduction: evolutionary and genetic features of albumin”. The paragraph devoted to comparative analysis of primary sequence of HSA, BSA and RSA has been moved from Chapter 1 to Chapter 2. The paragraph devoted to domain arrangement of albumin has been from Chapter 1 to Chapter 2. The paragraph devoted to interaction of albumin with endogenous and exogenous ligands has been transferred from Chapter 1 to Chapter 2.
Paragraph 8: the Authors have discussed the role of modified albumin in pathogenesis of diseases. Why did they discuss only the role of modified albumin? For example, several papers report the role of unmodified albumin and of its circulating levels in plasma in human diseases (e.g., cancer, infections). The rationale behind this choice is not clear.
We have not changed Chapter 8, instead we have expanded Chapter 9, which is devoted to integrative properties of albumin in diagnostics and therapy:
Several papers report the role of albumin level in plasma and different human diseases such as infections [Kaur t al., 2015; Yang et al., 2016; ], cancer [Merriel et al., 2016; Ikeda et al., 2017] and even depression in HIV-infected patients [Poudel-Tandukar et al., 2017].
Ikeda S, Yoshioka H, Ikeo S, et al. Serum albumin level as a potential marker for deciding chemotherapy or best supportive care in elderly, advanced non-small cell lung cancer patients with poor performance status. BMC Cancer. 2017;17(1):797. Published 2017 Nov 28. doi:10.1186/s12885-017-3814-3.
Kaur N, Kaur N, Sarangal V. A study to evaluate the correlation of serum albumin levels with chronic periodontitis. Indian J Dent Res. 2015;26(1):11-14. doi:10.4103/0970-9290.156788.
Merriel SW, Carroll R, Hamilton F, Hamilton W. Association between unexplained hypoalbuminaemia and new cancer diagnoses in UK primary care patients. Fam Pract. 2016;33(5):449-452. doi:10.1093/fampra/cmw051.
Poudel-Tandukar K, Jacelon CS, Bertone-Johnson ER, Palmer PH, Poudel KC. Serum albumin levels and depression in people living with Human Immunodeficiency Virus infection: a cross-sectional study. J Psychosom Res. 2017;101:38-43. doi:10.1016/j.jpsychores.2017.08.005.
Yang C, Liu Z, Tian M, et al. Relationship Between Serum Albumin Levels and Infections in Newborn Late Preterm Infants. Med Sci Monit. 2016;22:92-98.
Figure 6: the meaning of this figure is not clear. The caption of the figure indicates “Albumin and endothelial glycocalyx” but what should the reader comprehend from this figure? This figure illustrates the structure of the EGL, without providing any specific information on albumin.
The capture of Figure 6 has been expanded:
EGL serves as the "exclusion zone" or "gap" between endothelium and blood cells. The layer also acts as a filter, limiting the passage of the molecules larger than 70 kDa. Albumin with a molecular weight of about 66 kDa can firmly bind to EGL and penetrate to the interstitium by transcytosis.
The conclusions are not real conclusion, but a further paragraph. Indeed, the tile of this paragraph is not just “Conclusions” but “Conclusions: integrative properties of albumin in diagnostics and therapy.” This must be avoided, as conclusion should highlight the most important “take home message”.
Chapter 9 has been divided to two sections: Chapter 9. Integrative properties of albumin in diagnostics and therapy and 10.Conclusion.
Round 2
Reviewer 3 Report
The manuscript has been extensively revised according to the suggestion. The maniscript is now suitable fir publication?495”